# Identification of unique cell type responses in pancreatic islets to stress

Marlie M. Maestas[1,2], Matthew Ishahak [2], Punn Augsornworawat [3], Daniel A. Veronese-Paniagua[1,2], Kristina G. Maxwell [2,4], Leonardo Velazco-Cruz[1,2], Erica Marquez[2,4], Jiameng Sun[1,2], Mira Shunkarova[2], Sarah E. Gale[2], Fumihiko Urano [1,2,5] & Jeffrey R. Millman [1,2,4] ✉

Diabetes involves the death or dysfunction of pancreatic β-cells. Analysis of bulk sequencing from human samples and studies using in vitro and in vivo models suggest that endoplasmic reticulum and inflammatory signaling play an important role in diabetes progression. To better characterize cell type-specific stress response, we perform multiplexed single-cell RNA sequencing to define the transcriptional signature of primary human islet cells exposed to endoplasmic reticulum and inflammatory stress. Through comprehensive pair-wise analysis of stress responses across pancreatic endocrine and exocrine cell types, we define changes in gene expression for each cell type under different diabetes-associated stressors. We find that β-, α-, and ductal cells have the greatest transcriptional response. We utilize stem cell-derived islets to study islet health through the candidate gene *CIB1*, which was upregulated under stress in primary human islets. Our findings provide insights into cell type-specific responses to diabetes-associated stress and establish a resource to identify targets for diabetes therapeutics.

The primary function of the pancreatic islets of Langerhans is to regulate blood glucose. The β-cell, an endocrine cell within the islet, produces and secretes insulin to lower blood glucose levels. Failure of β-cells to properly maintain normoglycemia results in diabetes mellitus, an incurable metabolic disorder that affects hundreds of millions worldwide[1]. Disruptions to β-cell health can be caused by endoplasmic reticulum (ER) and inflammatory stress, which are connected to type 1 (T1D)[2–4] and type 2 diabetes (T2D)[5–8] onset and clinical progression. ER homeostasis is a balanced state between newly synthesized proteins entering the ER and properly folded proteins exiting the ER. ER stress occurs when there is an accumulation of unfolded and misfolded proteins in the ER. This can be caused by a multitude of reasons in the pancreatic islet, including inflammatory cytokines, high glucose, and free fatty acids[7]. In all cell types, ER stress is regulated by the unfolded

protein response (UPR) pathway[9]. Physiological and transient activation of the UPR is necessary for proper insulin processing in the ER, but pathologically high and chronic levels can lead to activation of cell death pathways[10]. A better understanding of the transcriptional landscape of islets under ER and inflammatory stress conditions could lead to clinical treatments for diabetes.

Targeting UPR pathways or improving ER folding through genetic changes or chemical chaperones has restored normoglycemia in diabetic mouse models[11–13]. However, mouse models do not fully recapitulate human islets. Prior studies have also discovered compounds through high-throughput chemical screening that increased the survival of a β-cell line under exogenous stress compounds[14]. Bulk sequencing of primary human islets has described a Golgi stress signature[15]. However, bulk approaches on heterogeneous tissue (such

[1]Roy and Diana Vagelos Division of Biology and Biomedical Sciences, Washington University School of Medicine, MSC 8127-057-08, 660 South Euclid Avenue, St. Louis, MO 63110, USA. [2]Division of Endocrinology, Metabolism, and Lipid Research, Washington University School of Medicine, MSC 8127-057-08, St. Louis, USA. [3]Department of Immunology, Faculty of Medicine Siriraj Hospital, Mahidol University, Bangkok 10700, Thailand. [4]Department of Biomedical Engineering, Washington University in St. Louis, St. Louis, USA. [5]Department of Pathology and Immunology, Washington University School of Medicine, St. Louis, USA. ✉e-mail: jmillman@wustl.edu

as islets) mask cell-type-specific responses. Human pluripotent stem cell-derived islets (SC-islets) are a tool for the study and treatment of diabetes[16–19]. We and others have shown that genetically modifying SC-islets can reduce ER stress[20] and apoptosis[21]. These studies suggest that targeting ubiquitous regulators of the UPR and stress may be a promising approach for treating diabetes. However, the specific transcriptional regulators of cellular stress response in the various primary islet cell types remain largely unknown.

Here, we examine the transcriptional responses to ER and inflammatory stressors[22–26] in isolated primary human islets using single-cell RNA sequencing (scRNAseq). Our analysis identified cell-type and tissue-specific stress response signatures and pathways, with β-, α-, and ductal cells being the most susceptible to stress. The comparison of β-cells to other pancreatic endocrine and exocrine cell types allowed for the identification of a β-cell specific stress signature. We modulated gene candidates in SC-islets to demonstrate the utility of this dataset to study islet health. These results provide a cell-type-specific resource of islet stress response, enhancing our understanding of islet health throughout the progression of diabetes.

## Results

### Human islet scRNAseq identifies stress-specific cell populations

To study cell-type-specific responses to ER and inflammatory stress, multiplexed scRNAseq was performed on three non-diabetic cadaveric

human islet donors (Supplementary Fig. 1a–i) and fixed scRNAseq on an additional two donors (Supplementary Fig. 2a–f) following ex vivo exposure to conditions that mimic diabetic stress (Fig. 1a). The islets were treated for 48 h with cytokines (IFNγ, TNFα, and/or IL1β), thapsigargin (TG), brefeldin A (BFA), or DMSO for the control (CTRL) (Supplementary Data 1). The cytokines can induce both ER stress[27] and inflammation. TG induces ER stress by inhibiting sarcoendoplasmic reticulum Ca$^{2+}$ ATPase (SERCA). BFA works by inhibiting the transport of protein from the ER to the Golgi causing ER and Gogi stress[28]. The treated cadaveric human islets were sequenced, and we ensured reproducibility by assessing each patient data individually. We found that across all five patients, the top differentially expressed genes had similar expression levels (Supplementary Fig. 1d–f, 2c, d, Supplementary Data 2–6). In addition, all five patients had similar proportions of cells across all treatment conditions, annotated cell type populations, and sequencing methods (Supplementary Fig. 3a–e). The treatment conditions in the hashed samples were identified through demultiplexing analysis (Fig. 1a, b, Supplementary Fig. 1h). The hashed samples were then integrated to minimize patient-specific findings (Fig. 1b, c).

We identified 14 clusters of cells based on the upregulation of canonical RNA markers, including α-, β-, δ-, Pancreatic Polypeptide (PP-), exocrine, immune, endothelial, and mesenchyme cells (Fig. 1c, d). There are four populations of α-cells (α-cell 1, α-cell 2, α-cell 3, and α-cell 4) and three populations of β-cells (β-cell 1, β-cell 2, and

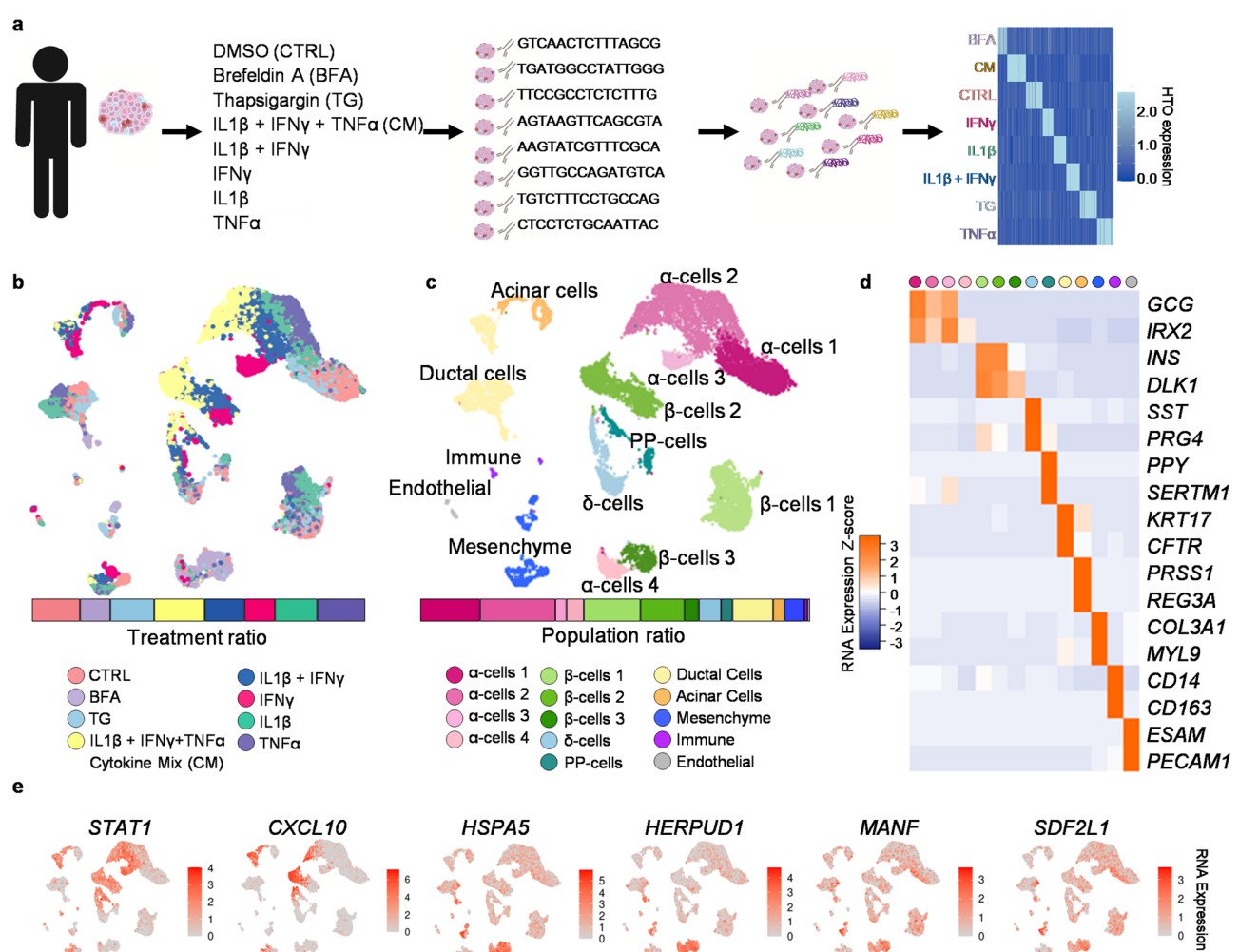

**Fig. 1 | Multiplexed single-cell sequencing of stressed primary human islets.**
**a** Schematic for multiplexing single-cell RNA sequencing of primary human islets with exogenous stressors. **b** UMAP showing which cells were treated with each stressor, and a bar graph of the ratio of each treatment in the entire population. **c** UMAP of islet cell populations based on RNA expression, and a bar graph of the ratio of cell types in the entire population. **d** Heatmap of key markers used to identify cell types in UMAP. **e** UMAP of key endoplasmic reticulum and inflammatory stress markers. Source data are provided as a Source Data file.

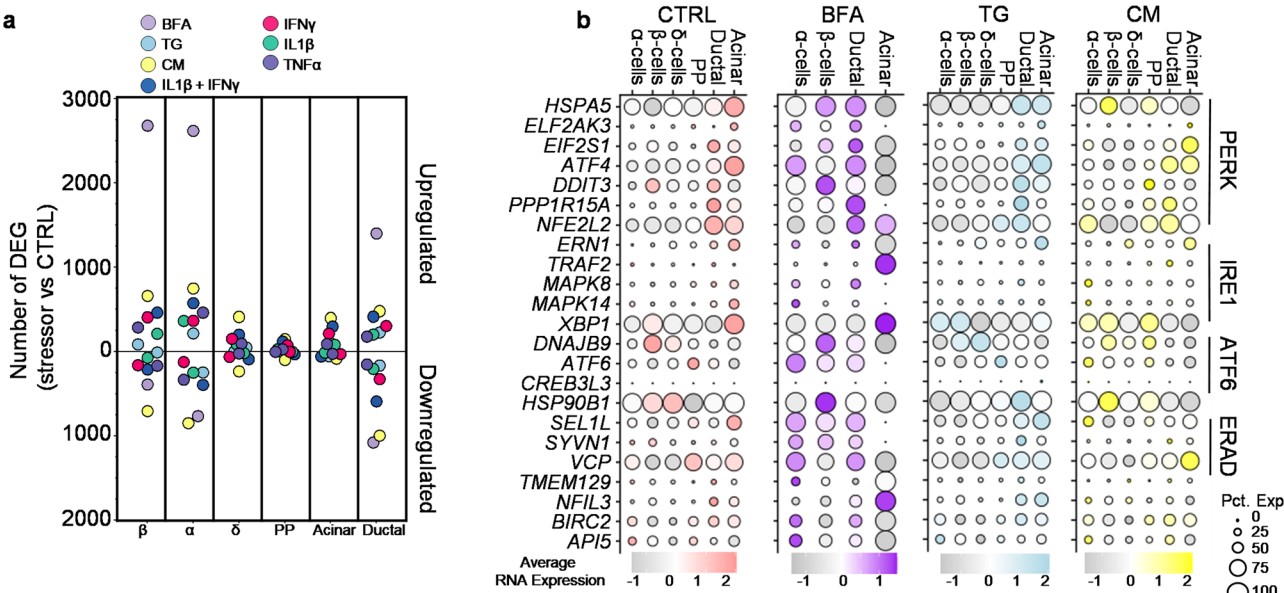

**Fig. 2 | Variable UPR regulation of stress response. a** Plot of differentially expressed genes (DEG) stressor vs. CTRL in endocrine and exocrine cell types. Above the zero are the number of upregulated genes and below the zero is the number of down-regulated genes. **b** Average RNA expression of known genes associated with the unfolded protein response (UPR) or ERAD under different stressors and across cell types. Circle size indicates the percent of cells expressing the gene. Source data are provided as a Source Data file.

β-cell 3). These sub-populations are not donor-specific and, instead, arise from the different exogenous stressors (Supplementary Fig. 1g, h). The α-cell 1 population is composed of CTRL, TG, and IL1β treated cells and expressed high levels of α-cell markers (Fig. 1b–d, Supplementary Fig. 1i). The α-cell 2 population consisted of cells treated with cytokine mix (CM, a combination of all three cytokines), IL1β + IFNγ, IL1β, and TNFα and showed increase expression of inflammatory markers. The α-cell 3 population had only IFNγ treated cells and showed upregulation of both inflammatory and α-cell markers. 85.1% of the cells found in the α-cell population 4 were treated with BFA, and these cells had decreased expression of α-cell markers and upregulation of UPR-associated genes (Supplementary Fig. 1i, Supplementary Data 7). Similar sub-populations of β-cells were observed, where the β-cell population 1 was treated with CTRL, TG, TNFα, and IL1β and had the highest expression of β-cell markers. In contrast, β-cell population 2 was treated with CM, IL1β + IFNγ, and IFNγ inducing increased inflammatory gene expression (Supplementary Fig. 1i). 94.6% of the cells in β-cell population 3 were treated with BFA and had decreased β-cell markers compared to other β-cell populations (Fig. 1d, Supplementary Fig. 1i, Supplementary Data 7).

To validate our stress conditions, we assessed the expression of genes from published literature[15,29,30] and found upregulation of *HERPUD1, HSPA5, COPZ2, GOLGA2*, GBF1, *CREB3*, and *COG2* under BFA stress; *MANF, SDF2L1, PPP1R15A, FKBP11, DDIT3, ATF4*, and *HERPUD1* under TG stress; and *STAT1, CXCL10, NOS2, RSAD2, ISG20, CD40*, and *OAS1* under CM stress (Fig. 1e, Supplementary Fig. 3f). This data was consistent across all five patients (Supplementary Fig. 3f). We validated our control cell population through comparative analysis with a published dataset of cadaveric human islets from donors without diabetes[31] (Supplementary Fig. 1j, k). We also conducted a Pearson correlation between hashed CTRL and fixed CTRL samples to confirm cell type populations between sequencing methods. We found the highest correlation between the same cell types across sequencing methods (Supplementary Fig. 3g, h). We also observed variation in the correlation value of matched cell types across cell types, which may be due to donor and processing variations. These data are consistent with

previous reports of these exogenous stressors[15,32] and validate our scRNAseq approach to studying human islets cultured under defined chemical stressors.

## Islet cell types have specific expression of UPR genes

We conducted a pair-wise analysis of differentially expressed genes (DEG) to determine cell-type-specific transcriptional changes due to stress. To broadly characterize the transcriptional divergence from the CTRL cells, we compared each stressor versus the CTRL in each major endocrine and exocrine cell type (Fig. 2a). BFA stress, which affects both the Golgi and ER[28], resulted in a robust transcriptional response with the most upregulated genes in β- (2679 genes), α- (2616 genes), and ductal cells (1399 genes). We found that cytokine mix also induces the most downregulated genes in β- (707 genes), α-(850 genes), and ductal cells (999 genes) (Fig. 2a). No δ-cells and PP-cells were observed within the BFA condition, so there is no DEG analysis for these cell types. Further, fewer DEG were observed in δ-, PP-, and acinar cells across other stress conditions compared to β-, α-, and ductal cells (Fig. 2a).

The UPR is a signaling network that works to mitigate ER stress[9]. During basal conditions, the islets show minor signs of UPR and ER stress as they are under constant pressure to produce hormones that regulate blood glucose levels[31]. We analyzed the three major UPR and endoplasmic reticulum-associated protein degradation (ERAD) pathways across cell types and stressors. Under BFA stress, β-cells have the highest expression of the pro-apoptotic gene, *DDIT3*. (Fig. 2b). Conversely, BFA stressed α- and ductal cells have higher expression of *BIRC2* and *API5*, pro-survival genes[33], compared to BFA-treated β-cells. The three stress sensors that regulate the UPR are PERK, IRE1, and ATF6. In ductal and acinar cells, CM and TG exposure induced the highest expression of PERK pathway-associated genes. In contrast, endocrine cell types responded to these stressors by increased expression of genes associated with the ATF6 pathway (Fig. 2b). Across stressors, we detected very few cells with IRE1 pathway-associated gene expression. In addition to the UPR, ERAD plays a role in preserving homeostasis in the ER[34]. This pathway is expressed highly under

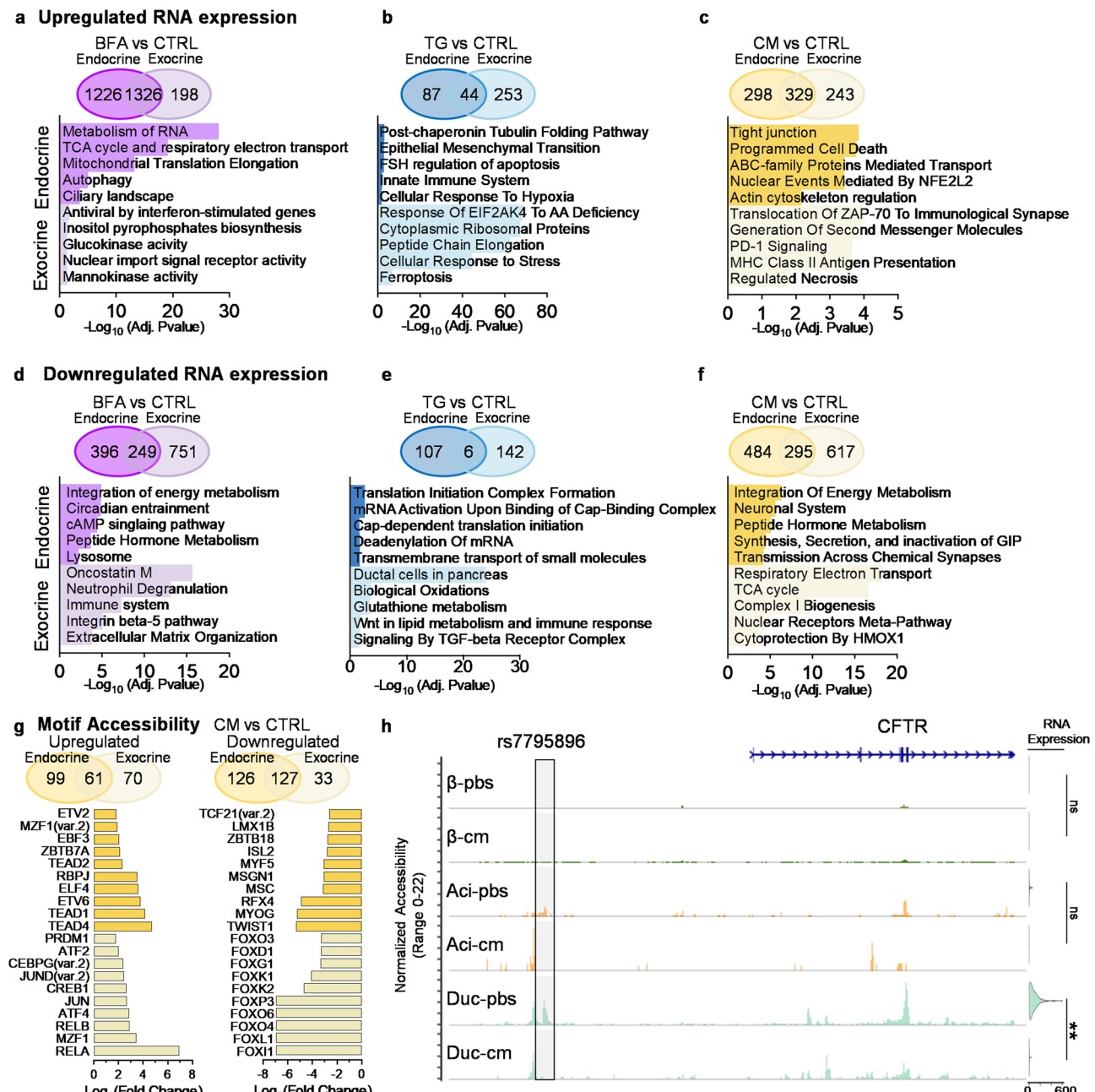

**Fig. 3 | Tissue-specific differences in response to stress. a–f** Differential gene expression of stressor vs control. Compared endocrine (β-, α-, δ-, PP-cells) vs exocrine (Ductal and Acinar) in Venn diagram. Below the Venn diagram is a bar graph using EnrichR analysis. The top 5 pathways are specific to endocrine, and the bottom 5 are specific to exocrine, (**a**) upregulated genes in BFA vs CTRL, (**b**) upregulated genes in TG vs CTRL, (**c**) upregulated genes in CM vs CTRL, (**d**) downregulated genes in BFA vs CTRL, (**e**) downregulated genes in TG vs CTRL, and (**f**) downregulated genes in CM vs CTRL. DEG cutoff was >0.25 Log₂(fold change) for upregulated or < −0.25 Log₂(fold change) for downregulated and <0.05 adjusted P-value. EnrichR uses the Benjamini-Hochberg method to correct for multiple hypotheses testing. **g** Motif accessibility of CM treated cells vs CTRL. Venn diagram compares endocrine (β-, α-, δ-, PP-cells) vs Exocrine (Ductal and Acinar). The left Venn diagram and bar graph are upregulated motif accessibility, and the right is downregulated. The bar graph shows the Log₂(fold change) of motifs specific to endocrine (top 10) or exocrine (bottom 10). **h** Coverage plot of *CFTR* accessibility and violin plot of RNA expression across endocrine and exocrine cell types. Adjusted *P*-value **\*\*P* = 3.11 × 10⁻³¹ or ns for non-significant. Statistical significance was determined by a two-sided Wilcoxon rank sum test. Highlighted is risk variant rs7795896. Source data are provided as a Source Data file.

BFA stress in endocrine and ductal cells, whereas it is higher in ductal and acinar cells under TG stress. The individual cytokines showed mostly upregulation of PERK and ERAD pathways in ductal and acinar cells, while α-, β-, δ-, and PP-cells are more variable across UPR and ERAD pathways (Supplementary Fig. 4a–d). Taken together, our analysis reveals cell-type-specific transcriptional changes associated with the UPR and ERAD pathways in various endocrine and exocrine cell types under different stress conditions.

**Stress-induced changes in gene expression are tissue-specific**
Endocrine cells, which include α-, β-, δ-, and PP-cells, are identified by *CHGA* and *CPE* expression (Supplementary Fig. 5a). Exocrine cells, including acinar and ductal cells, are identified by *KRT7* and *KRT19* expression (Supplementary Fig. 5b). We compared the upregulated of DEGs between BFA and CTRL and found 1326 genes shared between tissue types (Fig. 3a, Supplementary Data 8). The 1226 genes specific to endocrine are statistically associated with pathways involved in the

metabolism of RNA (R-HSA-8953854), the tricarboxylic acid cycle (TCA) (BioPlanet_2019), autophagy (R-HSA-9612973), and the ciliary landscape (WP4352). In contrast, the 198 upregulated genes specific to exocrine are involved with mannokinase/glucokinase activity (GO:0019158, GO:0004340), antiviral by interferon-stimulation (Bio-Planet_2019), and inositol pyrophosphates biosynthesis (PWY_6369) (Fig. 3a, Supplementary Data 8, 9).

Islets treated with TG had 44 shared genes upregulated in endocrine and exocrine cells (Fig. 3b). The endocrine cells specifically had genes related to tubulin folding (R-HSA-389977), epithelial-mesenchymal transition (MsigDB_Hallmark_2022), innate immune system (R-HSA-168249), and hypoxia/apoptosis (MsigDB_Hallmark_2022); the exocrine cells had genes involved in response of EIF2AK4 to amino acid deficiency (R-HSA-9633012), cytoplasmic ribosomal proteins (WP477), and ferroptosis (WP4313). TG induces a larger effect on DEG in exocrine than endocrine (Fig. 3b, Supplementary Data 8, 9).

Interestingly, exocrine cells that have been treated with CM showed upregulation of genes associated with PD-1 (R-HSA-389948) and T1D (KEGG_2021) (Fig. 3c). Genes that are upregulated are part of the major histone complex II (MHC Class II), including *HLA-DRB5, HLA-DQA2, HLA-DQA1*, and *HLA-DPA1* (Fig. 3c, Supplementary Data 8, 9). These genes are also upregulated within exocrine tissue treated with IL1β + IFNγ and IFNγ; however, IL1β alone and TNFα treated cells do not have these genes upregulated (Supplementary Fig. 5c–j, Supplementary Data 8, 9). Upregulated genes in endocrine are associated with programmed cell death (R-HSA-5357801) and actin cytoskeleton regulation (BioPlanet_2019) (Fig. 3c, Supplementary Data 8, 9). Overall there is a similar number of upregulated genes between endocrine and exocrine in cells treated with IL1β + IFNγ, IL1β, and IFNγ (Supplementary Fig. 5c–e). However, TNFα induces a more significant response in endocrine cells than exocrine (Supplementary Fig. 5i). In addition, IL1β induces upregulation of genes related to T cell receptor regulation of apoptosis (Reactome_2022) in both endocrine and exocrine tissue (Supplementary Data 8, 9).

BFA induces downregulation of more genes in exocrine tissue compared to endocrine tissue (Fig. 3d). Pathways associated with these genes downregulated in exocrine tissue include the immune system (Reactome_2022) and neutrophil degranulation (R-HSA-6798695) (Supplementary Data 8, 9). However, genes downregulated in endocrine tissue are related to the integration of energy metabolism (Reactome_2022), circadian entrainment (KEGG_2021), and genes associated with lysosomes, which has previously been shown to be downregulated during ER stress[35]. In endocrine cells, TG stress stimulates the downregulation of genes related to the translational initiation complex formation (R-HSA-72649) and mRNA activation. At the same time, downregulated genes in exocrine cells exposed to TG stress are associated with glutathione metabolism (BioPlanet_2019) and Wnt in lipid metabolism and immune response (BioPlanet_2019) (Fig. 3e). Only six genes are downregulated under TG stress in both exocrine and endocrine tissue, *NR4A1, FOSB, DSP, SMIM24, KIF12*, and *TNFSF10* (Supplementary Data 8, 9). CM stress suppresses genes associated with energy metabolism (R-HSA-163685), peptide hormone metabolism (R-HSA-2980736), and transmission across chemical synapses (R-HSA-112315) in endocrine tissue (Fig. 3f). Exocrine tissue displayed decreases in TCA cycle (R-HSA-1428517) and respiratory electron transport (R-HSA-611105). Across different combinations of cytokines, we find that genes related to neuronal systems are commonly downregulated in endocrine tissue (Supplementary Fig. 5f–h, j). In summary, our analysis reveals distinct gene expression profiles and pathway associations between endocrine and exocrine cells under various stress conditions, highlighting the diverse molecular responses and possible cross-talk between tissue types during diabetic-associated stress.

We found upregulation of apoptotic-associated pathways in our gene ontology analysis, prompting us to investigate whether our CM cocktail induced cell death. We found no significant difference in the viability of cadaveric human islets when treated with CM compared to control (Supplementary Fig. 6a). We also wanted to determine if islets can recover from these inflammatory conditions. We treated cadaveric human islets with CM for 48 h and then removed and washed off the CM and found a reduction of *MT2A, CXCL11, CXCL9, IL32, SOD2, ISG15, SAA2*, and *LCN2* when compared to CM treated cells (Supplementary Fig. 6b). This recovery from treatment has been shown in other systems as well[35]. *MAFA* was reduced in islets treated with CM; however, after recovery, the cells were able to induce *MAFA* expression. This data indicates that human islets can recover after cytokine stress, even with significant transcriptional changes during inflammation.

## Chromatin accessibility of CM reveals similarities to T1D

To better understand the chromatin changes under stress, we treated cadaveric human islets with CM (IFNγ + TNFα + IL1β) or PBS + 1% bovine serum albumin (BSA) (PBS/CTRL). We then used single-nucleus multi-omic sequencing to obtain gene expression (RNA) and chromatin accessibility (ATAC) data from CM and PBS/CTRL-treated islets and identified cell types by RNA expression (Supplementary Fig. 7a, b) and promoter accessibility (Supplementary Fig. 7c). We found ATAC-defined cell clusters were composed of two distinct treatment groups, while RNA alone clusters are more homogenous between treatments (Supplementary Fig. 7a). We looked at genes that were upregulated in our single-cell sequencing data set, *CXCL10, SOD2, IL32, GBP4*, and *LY6E*, and found that the chromatin region associated with these genes in CM treated cells are also more accessible compared to PBS/CTRL cells (Supplementary Fig. 7d). We compared motif accessibility of endocrine (β-, α-, and δ-cells) and exocrine (ductal and acinar cells) cell types and found the top 20 accessible transcription factor-binding motifs in each cell type between CM and PBS/CTRL (Supplementary Fig. 7e). Across β-, α-, and δ-cells, the top 20 motifs for each cell-type share 3 motifs in common, FOSL1::JUND, RBPJ, and ZNF282, while within the exocrine population, they share 15 motifs as their top 20. When comparing endocrine and exocrine, the endocrine motifs are more ubiquitously upregulated across cell types, while the exocrine motifs are more specific to exocrine tissue (Supplementary Fig. 7e). We also find that more motif accessibility is downregulated in endocrine than upregulated (Fig. 3g). In patients with diabetes, studies have shown differential accessibility associated with T1D[36,37]. Here, we investigate the risk variant rs7795896, which has previously been shown to have lower accessibility in T1D patients and is associated with lower CFTR expression[36]. In our dataset, we find that the environmental factor of inflammation due to treatment with CM causes similar chromatin changes in ductal cells to that of a T1D patient (Fig. 3h). There was also a significant decrease in RNA expression of *CFTR* under CM treatment in ductal cells (Fig. 3h, Supplementary Data 8). These data support a similar cellular response to both genetic and environmental contributions to T1D.

## Stressed α-, β-, and δ-cells have distinct identities

To elucidate similarities and differences in stress response between the three main endocrine cell types (α-, β-, and δ-cells), we compared the DEG of each stressor versus the control (Fig. 4a). We found 2465 shared genes between α- and β-cells under BFA stress. Among the 2117 upregulated genes, *ARF4, CREB3*, and *COG6* are Golgi stress-related genes[15], and *ARL1* and *TTC3* are upregulated in patients with T2D[38] (Supplementary Data 10). Gene ontology analysis of shared genes under BFA stress revealed pathways related to ER-golgi (GO:0006888, GO:0048193, GO:0030134, GO:0012507, GO:0030127), which was expected due to the mechanisms of stress

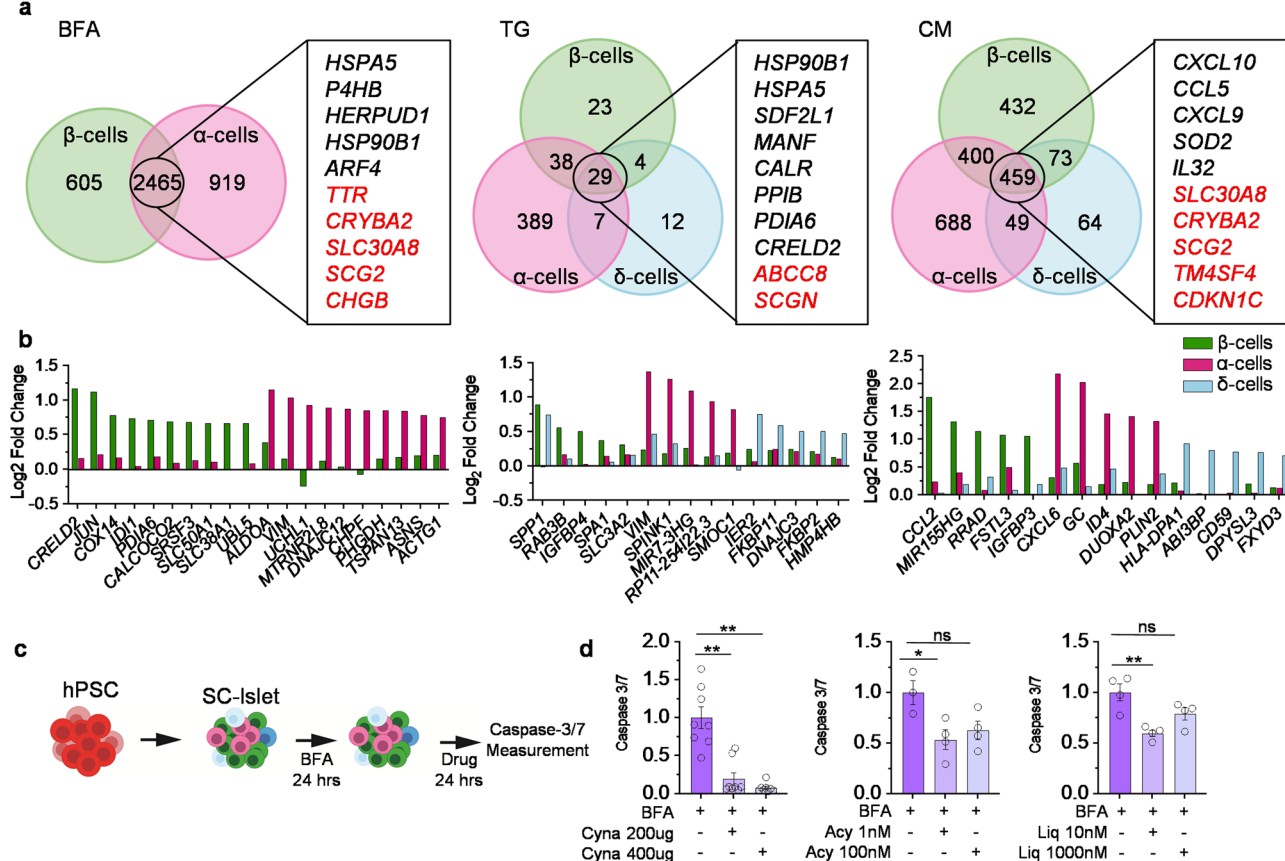

**Fig. 4 | Endocrine cell-type specific regulation of stress response. a** Venn Diagram of differentially expressed genes across β-, α-, and δ-cells. A list of genes that are shared across cell types, black is upregulated and red is downregulated genes. **b** Bar graph indicating Log$_2$(fold change) of cell-type specific gene regulation under stress conditions. **c** Stem cell-derived islets (SC-islets) treated with BFA for 24 h then the addition of a compound for 24 h. **d** Caspase 3/7 of SC islets with Cynaropicrin (Cyna) ($n = 8$, $P = 0.0001$), Acyclovir (Acy) (BFA, $n = 3$; Acy 1 nM, $n = 4$, $P = 0.024$) or Liquitirin (Liq) ($n = 4$, $P = 0.0026$). P-values were calculated by Dunnett's multiple comparison test. *$P < 0.05$, **$P < 0.01$ and error bar represent the s.e.m. Source data are provided as a Source Data file.

induced by BFA (Supplementary Data 11). Several genes associated with endocrine identity and insulin secretion (GO:0050796) were among the 286 downregulated genes shared between α- and β-cells.

Under TG stress 29 genes were differentially expressed in α-, β-, and δ-cells (Fig. 4a). *SCGN* and *ABCC8* are the only two genes downregulated across all three cell types, both of which have been correlated with T2D[39] (Supplementary Data 10). The three cytokines together induced an immune response, with 459 genes shared among the three cell types, 307 of which were upregulated and 152 downregulated (Fig. 4c). Upregulated genes shared between α-, β-, and δ-cells under CM stress are related to inflammation (GO:0050729, GO:0006954, GO:0050727) and immune response (GO:0034341, GO:0045088), while downregulated genes are associated with endocrine identity and ribosomes (GO:0042254, GO:0042274, GO:0042255) (Supplementary Data 11).

α-cells have more DEGs compared to β-cells under BFA stress. These genes encode proteins, including ALDOA (involved in glycolysis), VIM (involved in intermediate filaments), UCHL1 (found in neurons), and MTRNR2L8 (negative regulation of apoptosis). Gene ontology provided only two pathways specifically upregulated in α-cells and not β-cells; protein serine/threonine phosphatase activity (GO:0004722) has previously been implicated in other models of diabetes[40] and vesicles (GO:0031982). β-cells show upregulation of *CRELD2* (ER stress-responsive gene[41]), *JUN* (regulates gene expression), *COX14* (mitochondrial gene), and *IDI1* (up-stream of cholesterol synthesis) (Fig. 4b). Gene ontology of upregulated genes in β-cells are related to double-stranded RNA binding. Recent data has indicated disruptions to RNA editing can upregulate double-stranded RNA

leading to increases in inflammatory response and islet cell death[42]. Downregulated genes in β-cells are associated with metabolic pathways (GO:0031966, GO:0005747, GO:0005761, GO:0045333). Metabolism is essential for the proper function of pancreatic β-cells, and metabolic changes could result in differences in post-translational modifications that are related to T1D[43].

TG induces almost 17 times as many genes to be differentially expressed in α-cells than β- or δ-cells. α-cells also have the highest foldchange for cell-type specific genes. We see a similar trend under CM stress; however, a more subtle difference between the cell types is present. In β-cells, we find upregulation of *CCL2* (cytokine), *MIR155HG*, *RRAD* (GTPase transduction), *FSTL3* (glycoprotein), and *IGFBP3*. Only one significant gene ontology pathway, transition metal ion binding (GO:0046914), was related to genes upregulated under CM stress in β-cells compared to α- or δ-cells (Supplementary Data 11). We validated *IDI1*, upregulated under BFA stress, and *RAB3B*, upregulated under TG stress, using immunofluorescence (Supplementary Fig. 8a). *RRAD*, which is upregulated under CM stress, showed no apparent differences in protein levels between treatments. In α-cells, *CXCL6* (chemokine), *GC* (binds vitamin D), *ID4* (regulated gene expression), *DUOXA2* (ER protein), and *PLIN2* (coats intracellular lipid storage) are all upregulated under CM stress. In addition, gene ontology terms related to tight junctions (GO:0070160) and cadherin binding (GO:0045296) are downregulated in α-cells under CM stress. This could implicate a disruption to cell-to-cell contact during CM stress.

In CM, IL1β + IFNγ, IL1β, and TNFα, there is a significant downregulation of *CHGB*, *ABCC8*, and *PARVB* in all three cell types (Supplementary Fig. 8b–e, Supplementary Data 10). IFNγ induced the

most transcriptional changes in β-cells compared to the other individual cytokines, which has previously been reported[44]. To better understand the transcriptional response of each cytokine, we compared each cytokine combination to each other. We found 13 upregulated but no downregulated genes shared in α-, β-, and δ-cells across all five combinations of inflammatory stressors (Supplementary Fig. 8f, g). Most are directly related to the immune system and inflammatory response, while NCOA7 regulates RNA polymerase II, TYMP promotes angiogenesis, and GLRX is a member of the thioredoxin family (Supplementary Fig. 8f, Supplementary Data 10). We next compared cell-type responses to specific signaling pathways related to each cytokine. We found that under control conditions, α-cells seem to have the highest expression of genes associated with antigen processing and cross-presentation, signaling by interleukins, interferon gamma signaling, and TNFR2 non-canonical NF-kB pathway (Reactome_2022) (Supplementary Fig. 8h). As previously reported, α-cells have a high expression of HLA-E, an immune-inhibitory molecule, under different cytokine stress[45,46]. However, we find more heterogeneity across cell-types under stress conditions (Supplementary Fig. 8h). In antigen processing and cross-presentation, we see a switch from α-cells in CTRL conditions having high expression of genes to IL1β + IFNγ treated β-cells having a higher expression of genes associated with CD8 + T cells and promote their infiltration, such as PSMB8-10 and PSME2[47].

One of the major processes by which β-cells die is through apoptosis[48]. To study apoptosis, we generated SC-islets[49,50] and determined that BFA induces high levels of caspase-3/7, reduces expression of islet cell identity markers, and increases UPR-associated gene expression (Supplementary Fig. 9a-c), similar to our results from the cadaveric human islet scRNAseq data. We explored drug-gene interactions using The Drug Gene Interaction Database (DGIdb)[51] and DrugBank to establish the possible utility of this dataset for drug discovery. Three candidate genes RELA (subunit NF-KB), CKB (enzyme to transfer phosphates), and P4HB (part of oxidoreductase complex), were all upregulated under BFA stress in α- and β-cells (Fig. 4a). Using the databases, the corresponding drugs are Cynaropicrin, an inhibitor of RELA[52], Acyclovir, which is an antiviral agent previously shown to reduce insulin resistance in a mouse model[53], and Liquiritin, which has shown anti-inflammatory effects[54]. To test the impact on caspase 3/7, we treated our SC-islets with BFA for 24 h and then added each drug at different concentrations for an additional 24 h (Fig. 4c). We found all three drugs reduced caspase 3/7 in SC-islets under BFA stress (Fig. 4d). These data reveal the utility of this scRNAseq data set to be leveraged to discover approaches to improve SC-islet health.

## Six genes upregulated in endocrine cell types under stress

Gene lists created from BFA, TG, or CM vs control in β-, α-, and δ-cells (Fig. 4a) were then narrowed down to include only the upregulated genes (Fig. 5a). As previously stated, 2165 genes are upregulated in β- and α-cells under BFA stress, 70 genes in at least two endocrine cell types under TG stress, and 530 under CM stress (Fig. 5a). This combined list of 2765 genes was then compared to find overlapping genes between stressors. Only six genes are upregulated under all three stressors: CIB1, ERP44, HSP90B1, NEAT1, SELK, and VMP1. Most of these genes are expressed at the highest in BFA-treated β- and α-cells, while expression in δ-cells is more variable between TG and CM (Fig. 5b). Protein expression of these genes varies across stressors, which could be due whole-islet differences (Supplementary Fig. 9d). However, HSP90B1 is upregulated under BFA and TG stress in primary islets. To understand the gene regulatory network (GRN) in which these six genes may be involved, we used single-cell regulatory network inference and clustering (SCENIC)[55] to identify possible transcription factor networks activating these genes (Fig. 5c). Here, we show the regulon activity of the top 10 GRNs that target at least three of the six genes. In the CEBPB(+) gene regulatory network, the gene CEBPB is increased in

diabetic animals[56] and the regulon targets ERP44, HSP90B1, NEAT1, and VMP1. JUNB(+) and JUND(+) are part of the AP1 transcription factor family and STAT1(+) inflammatory signaling. YY1(+) plays a role in histone modification, and the regulon activity is highest under BFA stress. CREM(+) is involved in cAMP-signaling transduction.

Calcium and integrin-binding protein 1 (CIB1) is a protein involved in many cellular processes[57]. However, its role in SC-islets has not been characterized. CIB1 is significantly upregulated in β- and α-cells under BFA, TG, CM, IL1β + IFNγ, IL1β, and TNFα (Supplementary Data 12). CIB1 has been shown to interact with a multitude of proteins[57]. When CIB1 binds with PRKDC, TBPL1, PSEN2, PPP3R1, PTK2, PAK1, PDK1, KCNN1, or ITGA2B, it activates these proteins, and these genes have increased expression in BFA-treated β-cells. However, the gene expression of proteins that CIB1 inhibits also increases in TG-treated cells (Fig. 5d).

To better understand the impact of CIB1 on islet health, we utilized SC-islets. We generated lentiviral short-hairpin RNA to knockdown (KD) or open-reading-frame to overexpress (OE) CIB1 during the terminal stage of SC-islet differentiation (Fig. 5e). We produced either a 3.48-fold reduction or 5.64-fold increase in CIB1 expression (Supplementary Fig. 10a). CIB1 KD resulted in significant increases in insulin secretion under both 2 mM glucose and 20 mM glucose. At the same time, CIB1 OE has no impact on insulin secretion (Fig. 5f, Supplementary Fig. 10b). Due to the functional changes in CIB1 KD SC-islets, we measured cytosolic calcium levels, which are regulated by glucose[58]. CIB1 KD increased cytosolic calcium, and CIB1 OE decreased the levels in SC-islets (Supplementary Fig. 10c). CIB1 KD increased the proinsulin/insulin ratio while decreasing insulin content (Supplementary Fig. 10d, e). CIB1 OE did not affect the proinsulin/insulin ratio but increased insulin content (Supplementary Fig. 10d, e). Flow cytometry confirmed a decrease in C-Peptide expression in CIB1 KD cells and no significant difference in CIB1 OE (Supplementary Fig. 10f). In addition, even though we saw upregulation of CIB1 in our sequencing data in all three cell types and differences in GCG expression after CIB1 KD or OE in SC-islets (Supplementary Fig. 11a), we see no significant differences in GCG or SST protein expression between CIB1 KD or CIB1 OE compared to control SC-islets (Supplementary Fig. 10f). We also find that CIB1 KD or CIB1 OE did not significantly change cell type proportions (Supplementary Fig. 10g). These data establish CIB1 as a regulator of β-cell function in vitro.

Next, we assessed the proliferation of CIB1 KD or OE under stress conditions. We found that both KD and OE induced reduction of Ki-67+ cells under TG stress. We see no significant difference between control and CIB1 KD in other stress conditions (Supplementary Fig. 10h, i). We also measured transcriptional changes following CIB1 KD or OE induced in homeostatic and stress conditions. Under basal conditions, CIB1 KD reduced INS and GCG and some UPR genes (ATF4, DDIT3, and PPP1R15A), while CIB1 OE increases GCG, HSPA5, and DDIT3 (Supplementary Fig. 11a). Under stress conditions, there is an increase in apoptosis under BFA, CM, and TG stress due to CIB1 KD (Fig. 5g), while CIB1 OE reduces apoptosis under BFA stress only. CIB1 KD significantly reduces TXNIP expression under CM stress. Yet, OE increases gene expression of islet identity markers and chaperones for the UPR (Supplementary Fig. 11b). Under TG stress, CIB1 KD reduces the expression of islet identity genes and most UPR genes but increases HSPA5 expression. In comparison, CIB1 OE increases expression of islet identity genes and UPR-associated genes (Supplementary Fig. 11c). BFA stress induces increases in INS and GCG expression, while decreasing TXNIP expression in CIB1 KD cells. Exposure to BFA in CIB1 OE cells induces increased expression of GCG, HSPA5, and MANF and decreases expression of TXNIP (Supplementary Fig. 11d); INS expression is also increased under these conditions, but it is insignificant. These data demonstrate that CIB1 has a role in controlling the expression of genes related to islet identity and UPR.

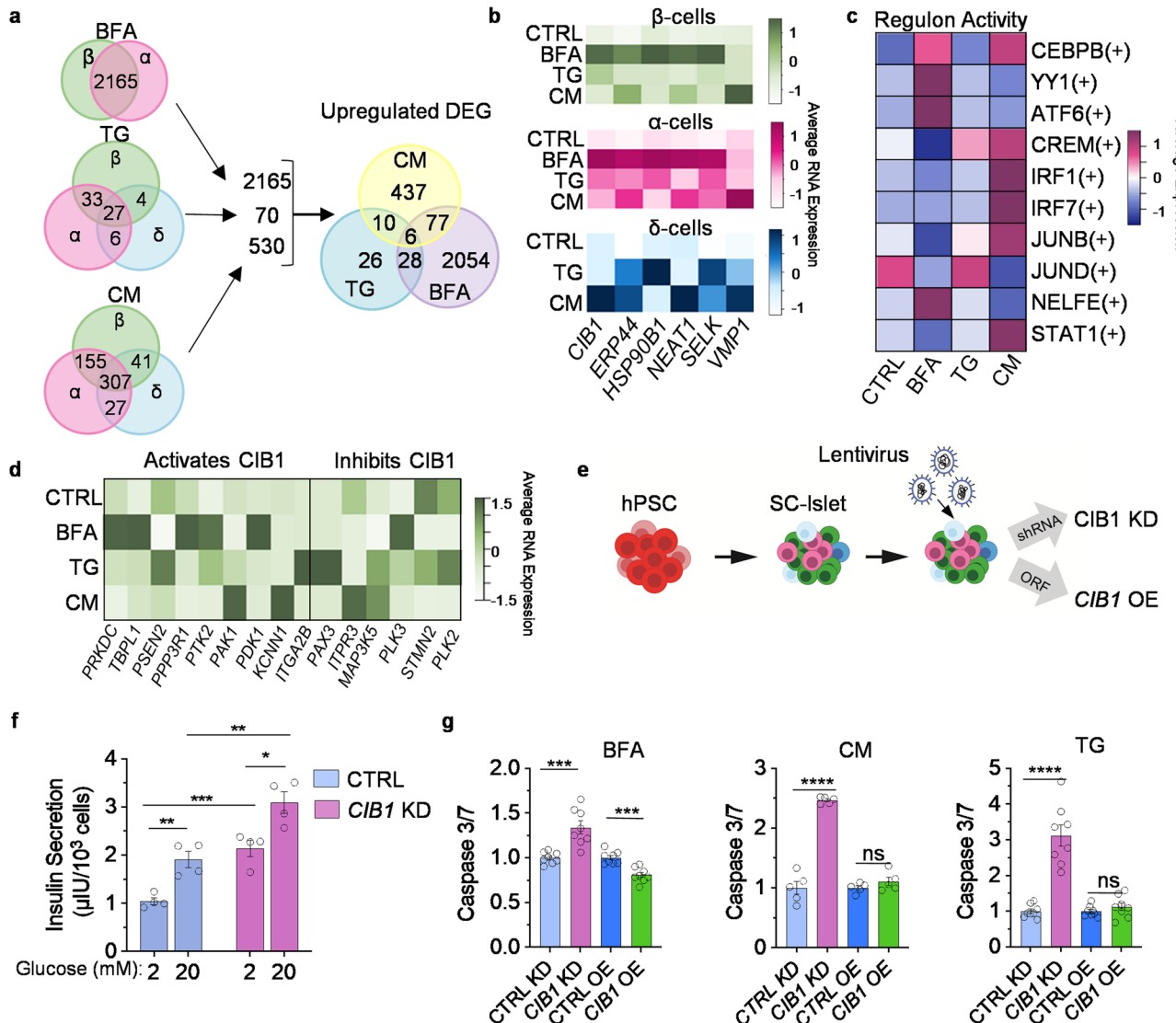

**Fig. 5 | Six genes are upregulated across endocrine cell types and stressors.**
**a** Upregulated DEG in β-, α-, and δ-cells across BFA, TG, and CM from 4a. Genes
that were in at least two cell types were used to make a new Venn diagram across
stressors. **b** Heatmap of the expression of the 6 upregulated shared genes in
CTRL, BFA, TG, and CM. **c** >3 of the shared 6 genes were found downstream of
these regulons in β-, α-, and δ-cells. **d** Heatmap of genes that activate/inhibit CIB1
in β-cells. **e** SC-islet differentiation with the addition of lentiviral short hairpin
RNA (shRNA) or open reading frame (ORF) targeting *CIB1*. **f** Static glucose-
stimulated insulin secretion of CTRL and *CIB1* KD ($n = 4$). The left two bars are

CTRL, and the right two bars are *CIB1 KD*. Paired $t$ test for 2 mM vs. 20 mM (CTRL
$P = 0.0067$, KD $P = 0.0249$), Unpaired $t$ test for 2 mM CTRL vs. 2 mM KD
($P = 0.001$) or 20 mM CTRL vs. 20 mM KD ($P = 0.0057$). **g** Caspase 3/7 assay of *CIB1*
KD and *CIB1* OE under stress conditions, BFA ($n = 8$; KD $P = 0.00048$; OE
$P = 0.000211$), CM ($n = 5$; KD $P = 1.1 \times 10^{-6}$), and TG ($n = 8$; KD $P = 6.87 \times 10^{-6}$),
unpaired $t$ test, **$p < 0.01$,***$p < 0.001$,****$p < 0.0001$, and not significant (ns) by
one-sided paired $t$ test. All error bars represent s.e.m. Source data are provided as
a Source Data file.

## Stress induces β-cell heterogeneity

During the progression of T1D, β-cells are preferentially attacked by
the immune system and are killed[48], while the other major endocrine
cell type, α-cells, have reduced glucagon secretion and gene
expression[59]. The reason for the different effects on islet cell types
during diabetes is unknown. Here, we define a β-cell-specific gene
expression signature under ER and inflammatory stress conditions by
comparing β-cells with other islet cell types (α-, δ-, PP-, acinar, and
ductal cells). We generated a subset of β-cells from all the other cell
types (Fig. 6a). Upon re-clustering, three predominate populations
arose, including a population composed of cells treated with IFNγ,
IL1β + IFNγ, and CM, a population of CTRL, TG, IL1β, and TNFα, and,
finally, a BFA population. The average expression of canonical β-cell
gene markers is decreased in BFA and CM stress (Fig. 6b). We also
found that BFA induces upregulation of proliferation and cell cycle

markers[60] in β- and α-cells, however, *MKI67* had the same expression
across all conditions (Fig. 6c, Supplementary Fig. 12a). We are uncer-
tain how much the lack of *MKI67* variation is caused by technical or
biological factors at this time. To determine changes specific to β-cells
under CM stress, we compared CM to control cells, in α-, β-, δ-, PP-,
acinar, and ductal cells to generate several gene lists. We then com-
bined these lists and delineated genes unique to β-cells (Supplemen-
tary Data 13). In total 657 genes are upregulated under CM stress in β-
cells, however, only 98 are unique to β-cells (Fig. 6d). These 98 upre-
gulated genes are statistically associated with xenobiotic metabolism,
TNF-alpha signaling via NF-kB, and interferon alpha response
(MsigDB_Hallmark_2020) (Supplementary Fig. 12b, Supplementary
Data 14). CM treated β-cells have 171 genes that are downregulated and
related to pancreas β-cells, DNA repair, protein secretion (MsigDB_
Hallmark_2020), and translation (R-HSA-72766) (Supplementary

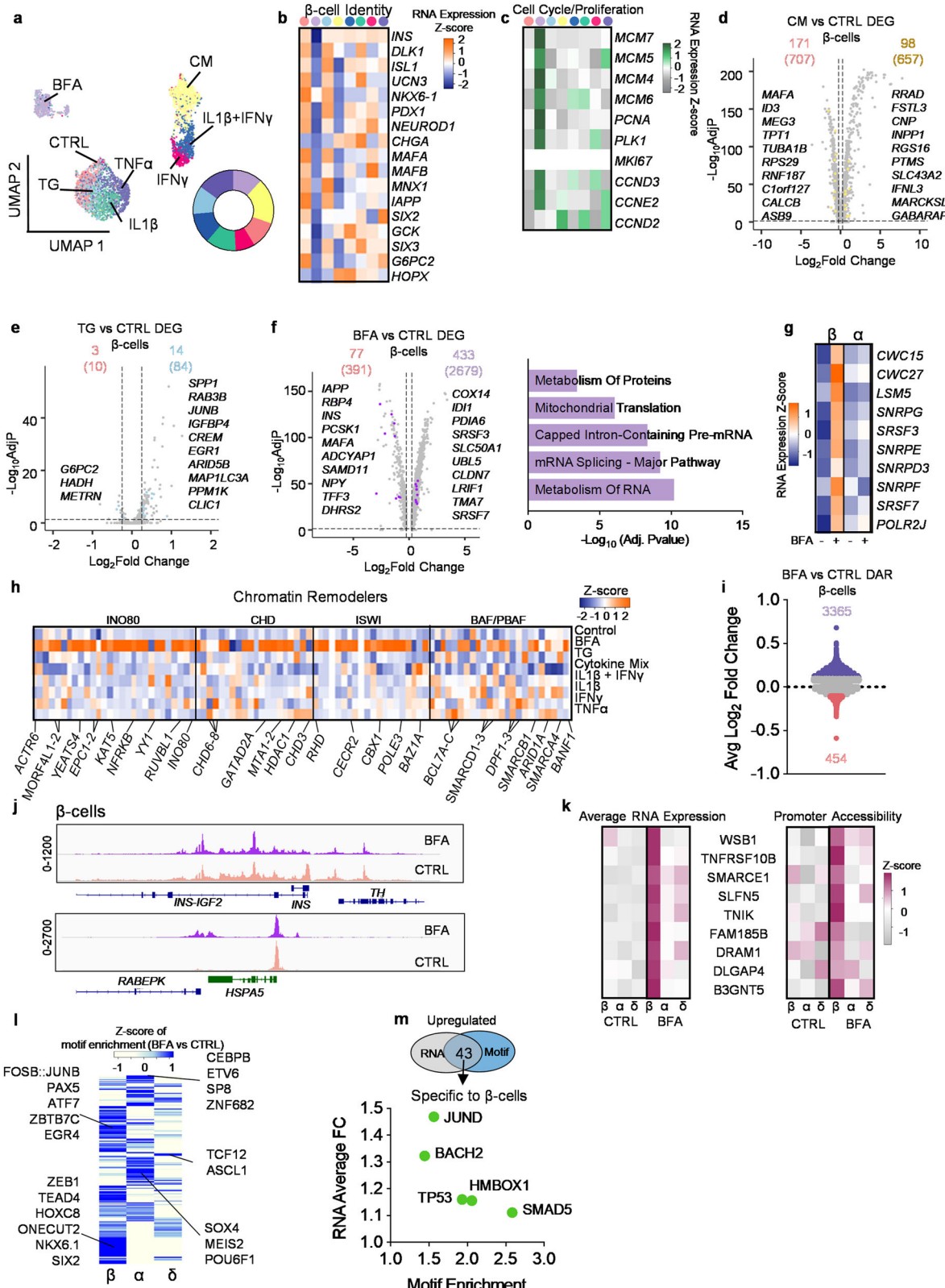

Fig. 12c, Supplementary Data 13,14). We also wanted to compare these results to α-cells and interestingly, upregulated genes specific to α-cells under CM stress are related to pathways involved in Myc targets V1, apoptosis, TGF-beta signaling, and heme metabolism (Supplementary Fig. 12d, Supplementary Data 15, 16), and downregulated genes are related to NMDA activation (R-HSA-438064, R-HSA-442755), integration of energy metabolism (R-HSA-163685) and neuronal

system (R-HSA-112316) (Supplementary Fig. 12e, Supplementary Data 15, 16).

Specifically in β-cells, TG-induced stress resulted in increased expression of *ARID5B* (part of demethylase complex), *MAP1LC3A* (Mitophagy), and *CLIC1* (chloride channel) expression and decreased *G6PC2* and *HADH*, which have been associated with diabetes (Fig. 6e, Supplementary Data 13). We could not conduct pathway analysis

**Fig. 6 | Stress-induced heterogeneity in the pancreatic β-cells. a** UMAP of only β-cells, circle is the proportion of stressors within the β-cell population (*n* = 4473). **b** Average RNA expression of known β-cell genes across stressors. **c** Heatmap of gene expression associated with cell cycle/ proliferation. **d–f**, Volcano plots of stressor vs CTRL. **d** CM vs CTRL, (**e**) TG vs CTRL, and (**f**) BFA vs CTRL. The top number is the number of genes specific to β-cells when compared to α-, δ-, PP, acinar, and ductal while the number in the parenthesis is the total number when FindMarkers is conducted. The parameters for genes were genes with a Log$_2$(fold change) >0.25 (upregulated) or Log$_2$(fold change) < −0.25 (downregulated) and adjusted *p*-value < 0.05. Statistical significance was determined by a two-sided Wilcoxon rank sum test. EnrichR was used for pathway analysis. **g** Average RNA expression of 10 genes associated with metabolism from (**f**). **h** Heatmap of average RNA expression of genes associated with chromatin remodelers across stressors in β-cells. **i** Differentially accessible regions in β-cells between BFA (top) and CTRL (bottom). **j** Comparing chromatin peaks near *INS, REBEPK*, and *HSPA5* in β-cells across BFA and CTRL. **k** Heatmap comparing β-cell-specific genes across cell types, stressors, and promoter accessibility. **l** Z-score of motif enrichment across β-, α-, and δ-cells. **m,** Venn diagram showing there are 43 overlapping upregulated genes and accessible motifs upregulated in β-cells. Dot plot are the 5 that are specific to β-cells. Source data are provided as a Source Data file.

because too few genes were differentially expressed between TG and CTRL. α-cells have 123 DEG under TG stress, including *PSMD8, PSMB6, PSMB3, PSMB1,* and *PSMA7*, which are involved in a variety of signaling pathways including Hedgehog (R-HSA-5358351, R-HSA-5362768, R-HSA-5387390, R-HSA-5610787) and cell cycle (R-HSA-69615, R-HSA-69615, R-HSA-69206, R-HSA-69656) (Supplementary Data 15,16). Genes downregulated in α-cells under TG stress are related to translation and the extracellular matrix (Reactome_2022) (Supplementary Fig. 12f).

Given that BFA stress induces a distinct sub-population, most DEGs are found in this condition, with 433 upregulated genes (Fig. 6f). Pathway enrichment shows these genes are associated with the metabolism of RNA, mRNA splicing, and processing of capped intron-containing pre-mRNA (Supplementary Data 14). Ten genes associated with these pathways are significantly upregulated only in β-cells and not α-cells under BFA stress (Fig. 6g). Downregulated genes specific to β-cells are associated with FOXO-mediated transcription (R-HSA-9614085), regulation of insulin secretion (R-HSA-422356), and β-cell development (R-HSA-186712). α-cells have upregulation of genes related to the dopamine neurotransmitter release cycle (Reactome_2022, Supplementary Data 15,16) and downregulated metabolism (R-HSA-1430728) and ion homeostasis (R-HSA-5578775). When comparing β-cell specific DEGs across all the cytokines, we found 0 upregulated genes that were differentially expressed in all the different combinations, however, all combinations significantly decreased expression of *ASB9, MAFA, NPTX2, RPS29, RNF187,* and *C1orf127* in β-cells (Supplementary Fig. 12g-i, Supplementary Data 13). We also compared our CM sequencing data with published T1D and AAB+ sequencing data[61] and performed pathway enrichment analysis across T1D, AAB+, and CM datasets using the upregulated DEGs. We found pathways related to antigen presentation and processing, cellular response to stress and stimuli, endosomal/vacuolar, and metabolism of proteins shared between T1D, AAB+, and CM (Supplementary Fig. 12j, Supplementary Data 17).

We were next interested in understanding if chromatin remodelers[62] are affected during ER stress in β-cells. Other model systems show the recruitment of remodelers after induction of ER stress[63]. Here, we found upregulation of genes associated with chromatin remodelers mainly under BFA-induced stress in β-cells (Fig. 6h). Given this finding, we conducted multi-omic sequencing on BFA and DMSO (CTRL) human islets to better understand chromatin and gene expression changes from the same cell. We integrated BFA and CTRL samples (Supplementary Fig. 13a). We used canonical RNA markers (Supplementary Fig. 13b) and promoter accessibility (Supplementary Fig. 13c) to define the cell types of the islet. We next assessed differentially accessible regions (DAR) between BFA and CTRL in β-cells. We found 8294 regions significantly (Adjusted *P*-value < 0.05) differentially accessible (Fig. 6i, Supplementary Data 18) and 3819 regions that had an absolute average Log$_2$(fold change) of at least 0.1. DAR associated with *INS* was significantly closed under BFA stress, while *RABEPK*, associated with endosomal trans-golgi network transport[64] was more open under BFA stress compared to CTRL (Fig. 6j). In α-cells, 4979 regions were significantly differentially accessible, with 2761

regions with an absolute average Log$_2$(fold change) of at least 0.1 (Supplementary Fig. 13d). In δ-cells, there were only 1216 DAR and 1043 met our threshold of 0.1 Log$_2$(fold change) (Supplementary Fig. 13e). We also compared RNA expression and promoter accessibility of genes that were significantly upregulated in β-cells but not α- or δ-cells (Fig. 6k). These genes included *WSB1*, a gene involved in hypoxia[65], *TNFRSF10B*, shown in other models to induce downstream inflammatory cytokine production after ER stress[66], and *SMARCE1*, a chromatin remodeler.

We used JASPAR2020[67] to identify motif accessibility across β-, α-, and δ-cells (Fig. 6l, Supplementary Data 18). Motif enrichment in β-cells shows an increase in β-cell identity, such as NKX6.1, SIX2, and ONECUT2[68]. To determine if the associated gene of the TF-binding motifs is also upregulated, we compared enriched motifs with upregulated genes under BFA stress in β-cells. We found 43 genes and motifs upregulated in β-cells (Fig. 6m). We then compared the list of 43 motifs to α- and δ-cell motif enrichment and found only 5 were unique to β-cells (Fig. 6m, Supplementary Data 18): *BACH2*, a risk gene for T1D[69]; *SMAD5*, a protein downstream of TGF-β signaling and can autonomously promote glycolysis[70]; *HMBOX1*, a regulator of telomerase activity[71]; *TP53*, a gene connected T1D and T2D by TP53-mediated apoptosis[72]; and *JUND*, a gene associated with β-cell dysfunction[73]. These data detail the many responses β-cells exhibit to diabetes-associated stresses.

## Discussion

Here, we comprehensively index transcriptional changes driven by diabetes-associated stress in pancreatic human islets. Previous studies have utilized cell lines, animal models, bulk sequencing, and ribosomal footprinting[35] to investigate exposure to cytokines[24,27,35,46,74–76] and ER stress[15,77] in the context of T1D. More recently, emerging single-cell technologies have led to insights using primary human islets and SC-islets[36,78–80]. Our use of single-cell/nuclei sequencing offers additional in-depth analysis of human cell-type-specific changes associated with ER and inflammatory stress. Future technological developments, such as single-cell proteomics, could be promising to further this study[81,82]. In addition, analysis of lipidomic, metabolomics, other post-translational modifications could advance our understanding of how ER and inflammatory stress drive diabetes progression[83–86]. Our findings also focus on 48 h treatments of human islets. At the same time, acute responses by reducing treatment timing or investigating chronic responses using long-term culture microphysiological systems[87–89] would also be interesting to examine in the future.

Our study revealed that diabetes-associated stressors induce cell-type-specific and tissue-specific transcriptional responses. Notably, stress-induced more DEG in α-cells than in β- or δ-cells. While the exact reason for this remains unknown, our dataset provides a resource to further investigate this observation. We found important differences in stress response between pancreatic endocrine and exocrine tissues. Surprisingly, stress downregulates more genes in exocrine tissue than in endocrine tissue. Endocrine cells have upregulation of genes associated with autophagy and programmed cell death across stressors. Exocrine cells have upregulation of MHC class II-associated genes

under CM stress, which previously has been shown to be enriched in ductal cells in patients with T1D[90]. These results provide some evidence of the influence of exocrine-endocrine crosstalk in diabetes pathology, a concept that is gaining increased attention[90–92].

Our ATAC data compares tissue types under CM stress and allows for the analysis of chromatin changes. Previously, T1D risk variant rs7795896 at the *CFTR* locus was identified to have lower accessibility[36], and here we found that CM stress also decreased accessibility in this region within exocrine tissue. Taken together, this suggests that both genetics and microenvironmental cues, like stress, could play a role in developing T1D via rs7795896. However, we could only secure one human islet donor for the ATAC data and, therefore, cannot speak to how common these findings are across donors. This study could be expanded upon by including a larger number of donors better representing the population, material from patients with diabetes, and a more comprehensive range of ages[68,90,93,94].

A recent paper by Chen et al. investigated ER stress in β-cells[35]. These authors found a specific signature comparing MIN6 cells to MEFs; our paper compared primary human β-cells with α-, δ-, acinar, and ductal cells. To further investigate the β-cell-specific signature, we conducted ATAC sequencing and found only five genes that were upregulated under BFA stress and had enriched motif accessibility. We did not further explore these five transcription factors in this study, but these results could be used in future experiments to study β-cell-specific stress responses. In addition, Chen et al. found loss of β-cell identity and the ability of the islet to recover from stress after recovery, which we also saw in our model system. However, they investigated translational differences, whereas in this study we focus mainly on transcriptional differences between islet cell types.

By regulating gene expression through genetic editing or the addition of a compound, we were able to alter the response of SC-islets to stress, establishing utility of this dataset. Prior studies have identified genes for islet survival, including genetic knockouts to prevent autoimmune rejection in mouse models[11,95], targeting the UPR to protect the islet[12–14,21], or granting human SC-islets various degrees of hypoimmunity or immunocloaking in vitro and in vivo[96–100]. In addition, compounds have been identified to reduce stress signatures found in cells with Wolfram Syndrome genetic variants[101,102]. Here, we discovered that *CIB1* is important for the function of SC-islets and response to apoptosis. One role CIB1 has is inhibiting protein-ligand of the inositol-1,4,5-triphosphate receptor (InsP3R)[57,103] which regulates intracellular calcium. *CIB1* KD might allow calcium release through InsP3R, resulting in increased cytosolic calcium and subsequently increasing the secretion of stored insulin, possibly explaining the change in function. We also identified several compounds that reduced caspase-3/7 under BFA stress in SC-islets in vitro. Compounds improving islet health could also be used to improve islet transplantation[104–108].

Our sequencing data provides cell-type-specific findings to study human islet health. A limitation of our study is that we were not able to source primary human islets for all components of the analysis. Another way to study islet health would be to generate SC-islets from patient cells by utilizing genetic engineering to introduce or correct disease-associated genetic variants[20,76,109–114]. Additionally, sequencing datasets from multiple sources could be combined into a more comprehensive atlas to reveal insights[115] to improve SC-islets for modeling, such as including endothelial cells[116–119]. Our robust scRNAseq of stressed primary islets provides a resource for future rigorous studies into identifying targets to improve diabetes treatments.

## Methods
Our research complies with all relevant ethical regulations. Non-diabetic islets were procured from Prodo Laboratories. Prodo Laboratories obtains informed consent that covers all non-identifiable information, which can be found at (https://prodolabs.com/human-islets-for-research), and compensation is not provided. These islets have been refused for transplantation, have been quality-controlled, and meet specific criteria for research purposes only. Sex was not considered in the study design because that was not within the scope of this study. The research in this study was approved by the Washington University Institutional Biological & Chemical (IBC) Safety Committee (Approval number 12186). Washington University Embryonic Stem Cell Research Oversight Committee approved all work utilizing HUES8 (Approval number 15-002).

### Cadaveric human islet processing
Cadaveric human islets from donors without diabetes were purchased from Prodo Labs. Cells were cultured in CMRL (Mediatech; 99-603-CV) + 10% FBS (GE Healthcare; SH3007003). Donor islets for sequencing comparisons are from patients as follows: (1: hashed) 30 yr male with BMI 22.7, (2: hashed) 65 yr female with BMI 25.1, (3: hashed) 56 yr male with BMI 24.3, (4: fixed) 64 yr male with BMI 25.5, (5: fixed) 47 yr male with BMI 28.3, (6: ATAC) 33 yr male with BMI 33.3.

### Stress treatment of islets
Human cadaveric islets were separately treated for 48 h under the following conditions: (1) DMSO (Fisher; BP231), (2) 1 ug/mL Brefeldin A (Sigma; B5936), (3) 10 μM Thapsigargin (Sigma; T9033), (4) Cytokine mix (1000 ng/mL IFNγ, 500 ng/mL TNFα, 100 ng/mL IL1β), (5) 100 ng/mL IL1β + 1000 ng/mL IFNγ, (6) 1000 ng/mL IFNγ (R&D; 285IF100), (7) 100 ng/mL IL1β (R&D; 201LB005), (8) 500 ng/mL TNFα (R&D; 210TA020). For multiomic sequencing, human islets were treated under the following conditions: (1) 24 h DMSO (CTRL), (2) 48 h PBS + 0.1%BSA, (3) 24 h 0.1 ug/mL Brefeldin A, (4) 48 h 1000 ng/mL IFNγ + 500 ng/mL TNFα + 100 ng/mL IL1β.

### Hashed single-cell RNA preparation and sequencing
Cells were prepared according to BioLegend TotalSeq A antibodies with 10X single cell 3' v 3.1 hashing protocol. Briefly, cells were single-cell dispersed using TrypLE and washed with PBS. Then 1-2 million cells were resuspended in 100 μl cell staining buffer (BioLegend 420201) and 10% FBS for 10 mins. Cells were incubated separately in 1 μg of hashed antibodies (BioLegend; 3946-01,-03,-05,-07,-09,-11,-13, -15) for 30 mins at 4 °C and washed twice with cell staining buffer. Finally, all cells were resuspended in DMEM at 1000 cells/μl and pooled together. These samples were processed by the McDonnell Genome Institute (MGI) at Washington University for library preparation and sequencing using the NovaSeq 6000 System (Illumina).

### Fixed single-cell RNA sequencing preparation
Patients 4 and 5 were sequenced using the 10X fixed RNA kit protocol. After treatment with stressors, cells were single-cell dispersed using TrypLE and washed with PBS. The cells were spun down and resuspended in fixation buffer (10X Genomics; PN-2000517) and 37% formaldehyde. The samples were incubated overnight for 20 h at 4 °C. Cells were resuspended in a quenching buffer (10X Genomics; PN-20000516) and counted using Countess II. Enhancer (10X Genomics; PN-20000482) was added to store samples at -80 °C. Once both patients were fixed, probe hybridization (Protocol 10x Genomics; CG000527) was conducted using 16 separate barcodes, 8 for patient 4 (Supplementary Data 5) and 8 for patient 5 (Supplementary Data 6). The samples were then pooled together and sent to Washington University in St. Louis MGI for sequencing.

### Single-cell RNA sequencing analysis
Datasets were analyzed using R version 4.0.3 and Seurat version 4.0. Quality control was conducted on each patient individually by excluding cells with high mitochondrial and RNA counts. Patient one (hashed sequencing) was filtered for unique feature counts between 2000 and 7500; patient two (hashed sequencing) was filtered between

1000 and 7500; and patient 3 (hashed sequencing) was filtered between 1000 and 6000. To demultiplex the hashed samples we used the Seurat function *HTODemux*. This function defines doublets as having more than one oligo barcode and a negative cell as having a low value for the oligo barcode. This uses a 0.99 quantile as a threshold to identify positive and negative cells. Only the positive cells were used for further analysis.

Then, the three hashed patients were individually LogNormalized using *Normalize Data*, and the top 2000 variable features were identified using *FindVariableFeatures*. *FindIntegrationAnchors* was used to identify anchors between the three patients. The anchors were then used to integrate the patients using *IntegrateData*. Following integration, *ScaleDate*, *RunPCA*, and *RunUMAP* were used on the integrated data set containing data from the three patients. Clustering and dimension reduction were done using dimensions of 1:20 and a resolution of 0.3. Pair-wise comparisons using *FindMarkers* and the Wilcoxon Rank Sum test were used to determine DEG with a $Log_2$ Fold change absolute value of $> |0.25|$ and adjusted $p$-value $< 0.05$ using Bonferroni correction. This threshold was set based on previously published data[31,35,46,78,79,90,120]. EnrichR[121–123] was used for gene set enrichment analysis. As input, we used gene lists specific to a cell type or tissue type separated by upregulated ($>0.25$ $Log_2$Fold Change) or downregulated ($<−0.25$ $Log_2$Fold Change) genes. We filtered the gene set enrichment analysis by an adjusted $p$-value $< 0.05$.

For the fixed samples, the different stress treatments were merged using the raw data counts. Then, patients 4 and 5 were filtered for 200-9000 genes in each cell and percent mitochondria below 5. *SCTransform* was used to normalize each patient and regressed out percent.mt. *RunUMAP*, *FindNeighbors*, and *FindClusters* were done on each patient. The dimensions of patient 4 clustering and dimensional reduction were 1:25 and a resolution of 1.6. The dimensions of patient 5 were 1:35 and a resolution of 1.6. To integrate the two fixed data sets, we used *SelectIntegrationFeatures*, with features set to 3000. Then, we used *FindIntegrationAnchors* and the integration features to find anchors. Next, *IntegrateData* was used to integrate data from the anchors. For UMAP projection, *RunUMAP* and *FindNeighbors* were set to a dimension of 1:25, and *FindClusters* at a resolution of 1.6.

To integrate fixed and hashed sequencing, each patient was Lognormalized using *NormalizeData*. Then *SelectIntegrationFeatures*, *FindIntegrationAnchors*, and *IntegrateData* was performed. The data was scaled, and *RunPCA* was performed. The dimensions for *RunUMAP* and *FindNeighbors* were set to 1:27, and *FindClusters* had a resolution of 1.5. The integration of fixed and hashed samples was used for Supplementary Fig. 3.

### Single-nuclei sequencing and preparation
Human islets were single-cell dispersed using TrypLE for 10 min at 37 °C and quantified for >90% viability using the Vi-Cell XR (Beckman Coulter). The 10X Multiome ATAC + Gene Expression (GEX) protocol (CG000338) was used to obtain the nuclei from the samples. The cell samples were collected and washed in PBS with 0.04% BSA, lysed with chilled lysis buffer for 4 mins, washed three times with wash buffer, and resuspended with 10x nuclei buffer at 3000–5000 nuclei/μl. Nucleus samples were processed using the Chromium 10x genomics instrument. The McDonnell Genome Institute at Washington University conducted library preparation according to the 10x Single Cell Multiome ATAC + Gene Expression v1 kit. The library was sequenced using the NovaSeq 6000 System (Illumina).

### Single-nuclei sequencing analysis
Cell Ranger ARC v2.0 was used on raw files. Genes were mapped and referenced to GRCh38, the human reference genome. RStudio 1.3.1093 (R version 4.0.3), Seurat 4.01, and Signac 1.3.0 were used for analysis. MACS2 was used to call peaks, and the genomic positions were mapped and annotated with EnsDb.Hsapeins.v86 and hg38. Low-quality

cells, including doublets, dead cells, and poor sequencing depth cells were removed by filtering out cells in BFA/DMSO with low RNA counts (nCount_RNA < 1000), and low ATAC counts (nCount_ATAC < 1000); high RNA counts (>40,000-50,000) and high ATAC counts (>40,000); nucleosome signal >1.5 and transcription start site (TSS) enrichment <2. In CM/PBS with low RNA counts (nCount_RNA < 1000) and low ATAC counts (nCount_ATAC < 1000), high RNA counts (>50,000), and high ATAC counts (>50,000-60,000), nucleosome signal >1.5 and transcription start site (TSS) enrichment <2. BFA was integrated with its control, DMSO, and CM with its control, PBS, using *SelectIntegrationFeatures*, *PrepSCTIntegration*, *FindIntegrationAnchors* normalizing to SCT from each dataset, and *IntegrateData*. To create merged ATAC data, we used *RunTFIDF* and *RunSVD*. To build an integrated UMAP of RNA and ATAC, we used *FindMultiModalNeighbors*. To add motif information, we used the JASPAR2020 database and genome BSgenome.Hsapiens.UCSC.hg38 to call *RunChromVar*. Promoter accessibility was found by calling *GeneActivity*. For differential motif expression, we used an adjusted $p$-value of <0.05 and average $log_2$Fold Change >0.1 or < -0.1 (Fig. 3g). *CoveragePlot* was used to graph rs7795896 peak accessibility. To show gene expression (SCT assay) and peak accessibility (peaks assay) on a UMAP we used *FeaturePlot*. Differentially accessible regions were found by using the "peaks" assay and *FindMarkers* comparing BFA and DMSO in β- or α-cells, logistic regression framework was used, and the variable to test was set to nCount_peaks.

### Stem cell-derived islet differentiation
The HUES8 stem cell line was generously provided by Dr. Douglas Melton (Harvard University). HUES8 stem cell line was differentiated using our published method[49,50]. Undifferentiated hPSCs were seeded onto a Matrigel (Corning; 354277) treated tissue culture flask at 0.63 × $10^6$ cells cm$^{-2}$ with mTesR1 (StemCell Technologies; 05850) and 10 μM Y-27632 (Pepro Tech; 129382310MG). After 24 h, media and growth factors were added to start the differentiation process (Supplementary Data 1). On Stage 6 Day 7, the differentiation was dispersed with TrypLE at 0.2 ml cm$^{-2}$ (Gibco; 12-604-013) and seeded into individual wells of a 6-well dish with 4-mL of ESFM media on an Orbi-Shaker (Benchmark) at 100 RPM. After 6-12 more days, cells were used for assessment. All factors, timing, and media forumulations for the differentiation are in Supplementary Data 1.

### Stress treatment SC-islets
SC-islets were treated with (1) Control (DMSO), (2) 24 h 0.1 ug/mL Brefeldin A, (3) 48 h 10 μM Thapsigargin, (4) 48 h Cytokine mix (500 ng/mL IFNγ, 500 ng/mL TNFα, 100 ng/mL IL1β).

### Transduction of lentiviral gene-editing
Lentiviral transduction was initiated on Stage 6 Day 7 during aggregation with a multiplicity of infection (MOI) of 5 for 24 h. Plasmids containing sequences targeting the gene of interest or GFP (Control) were ordered. Plasmid DNA was isolated using a QIAprep Miniprep kit (Qiagen; 27115) and then transformed into One Shot™ Stbl3™ Chemically Competent *Escherichia coli* (Invitrogen; C737303). Single colonies were selected, cultured, and DNA was extracted using Qiagen Maxi plus kit (Qiagen; 12981). Viral particles were generated using Lenti-X 293T cell line (Takara; 632180) and cultured in DMEM + 10% heat-inactivated Fetal Bovine Serum (MilliporeSigma; F4135) + 0.01 mM Sodium Pyruvate (Corning; 25-000-CL) in 10-cm tissue culture treated plates (Falcon; 353003). Next, Lenti-X 293 T cells were transfected with 6-μg of plasmid DNA, 4.5 μg of psPAX2 (Addgene; 12260), 1.5 μg pMD2.G (Addgene; 12259), and 48 μL of Polyethylenimine Max (Polysciences; 24765-2). Viral supernatant was collected at 96 h post-transfection and concentrated using Lenti-X concentrator (Takara; 631232). The lentivirus was titered using Lenti-X™ GoStix™ Plus (Takara;631280). MOI of 5 is used for all viruses.

## Caspase 3/7 Assay

On stage 6 day 7 of SC-islet differentiation, cells were transduced with lentivirus to knock down or overexpress *CIB1*. On Stage 6 Day 12 cells were single-cell dispersed and plated onto Matrigel in 96-well black plates at 15,000 cells per well. After 24 h, cells were treated with stressors, BFA (0.1 ug/mL), TG (10 uM), or CM (IL1B 100 ng/mL + IFNy 500 ng/mL + TNFa 500 ng/mL). After 48 h (TG, CM) or 24 h (BFA), we measured Caspase-3/7 according to the manufacturer's instructions (Promega, G8093). For compound treatment, cells were single-cell dispersed and plated on Stage 6 Day 12. After 24 h, cells were treated with BFA (0.1 ug/mL). After an additional 24 h, the compound was added at the following concentrations: (1) Cynaropicrin (200 ug/mL or 400 ug/mL), (2) Acyclovir (1 nM or 100 nM), (3) Liquitirin (10 nM or 1000 nM) for an additional 24 h. After another 24 h, the assay was conducted.

## Quantitative real-time PCR (q-PCR)

Cell clusters were collected, washed, and resuspended in an RLT buffer. RNA was extracted from SC-islets using the RNesy Mini Kit (QIAGEN; 74016) and DNase treatment (QIAGEN; 79254). cDNA was then synthesized using the High-Capacity cDNA Reverse Transcriptase Kit (Applied Biosystems; 4368814) and a T100 thermocycler (Bio-Rad). PowerUp SYBR Green Master Mix (Applied Biosystems; A25741) generated real-time PCR reactions on the QuantStudio™ 6 Pro System. Data was analyzed using the DDCt methodology. Normalization markers used were TBP and GUSB. Primer sequences are available in Supplementary Data 1.

## Proinsulin and insulin content

Proinsulin to insulin content was measured by collecting SC-islets, rinsing with PBS, and incubating in an acid–ethanol solution for 48 h at −20 °C. Samples were neutralized with 1 M Tris buffer (Millipore Sigma; T6066). Enzyme-linked immunosorbent assay (ELISA) kits: human insulin ELISA (ALPCO; 80-INSHU-E01.1) and human pro-insulin ELISA (Mercodia; 10-1118-01) were used to measure protein levels. Results were normalized to cell counts performed on Vi-Cell XR (Beckman Coulter).

## Glucose-stimulated insulin secretion (GSIS)

Static GSIS was conducted using stage 6 day 14 SC-islet clusters. The clusters were placed in a transwell (Corning) and washed with 1 mL of Krebs-Ringer Bicarbonate (KREB buffer, Supplementary Data 1) three times. Then the transwells were transferred to 2 mM glucose for 1 h to equilibrate cells. Next, cells were moved into 2 mM glucose for 1 h and 20 mM for 1 h, and the supernatant was collected for both 2 mM and 20 mM.

## Cytosolic calcium assay

Calcium was measured in Stage 6 Day 14 SC-islets following the Fura-2AM protocol (ab176766). Cells were single-cell dispersed and plated at a density of 60,000 cells/well onto Matrigel onto a 96-well plate. 20 uL of Fura-2 AM stock was added to 10 mL 1X assay buffer, then 100 uL was added to each well of cells. Cells were incubated for 1 h at 37 °C, then 20 min at room temperature. Fluorescence was measured at an excitation of 340 nm and 380 nm and an emission of 510 nm. Finally, 340/380 ratios were calculated.

## Immunocytochemistry

Samples were plated on Matrigel-coated 96 black plates after lentiviral transduction. After 24 h, cells were treated with BFA, TG, or CM (described in the stress treatment section). Cells were then fixed with 4% paraformaldehyde for 30 min at room temperature. Samples were then blocked for 45 min at room temperature with PBS + 0.1% Triton-X 100 + 5% donkey serum. The primary antibody was incubated

overnight at 4 °C. The secondary incubated the next day for two hours at room temperature. DAPI was used to stain nuclei. Imaging was performed on a Leica. Quantification of fluorescence was done using ImageJ. Antibody details can be found in Supplementary Data 1.

## Immunohistochemistry

Stress-treated human islets were paraffin-embedded. To remove paraffin slides were put into histoclear for 10 min, 100% EtOH 4 min, 95% EtOH 4 min, 70% EtOH 4 min, and then rinsed with DI water. Antigen retrieval was performed using 0.05-0.1 M Ethylenediaminetetraacetic acid (EDTA) for 2 h in a pressure cooker. Slides were blocked with PBS + 0.1% Triton-X 100 + 5% donkey serum for 30 min at room temperature. Primary antibody was added at a 1:200 dilution, overnight at 4 °C. Samples were incubated in secondary for 2 h at room temperature (1:300 dilution). Slides were mounted using DAPI Fluoromount-G. Leica was used for imaging.

## Flow cytometry

SC-islets were transduced with short-hairpin RNA GFP (CTRL), shCIB1, Open-reading frame GFP (CTRL), or CIB1 OE. After 6 days, cells were collected and dispersed using TrypLE. The cells were fixed with 4% paraformaldehyde aqueous solution (PFA, 157-4-100) for 20 min at RT. Then PFA was removed, and cells were washed with PBS. Blocking and permeabilized were done using 5% donkey serum and 0.1% Triton-X in PBS for 30 min on ice. Cells were then incubated in primary antibody overnight at 4 °C (C-peptide 1:300 DSHB GN-ID4-S, GCG 1:350 BD565891, SST 1:250 BD566032). Next day secondary was added and incubated for 2 h in the dark. Cells were filtered and Cytek Northern Lights was used for flow cytometry. Analysis was done using FlowJo.

## Western blots

Primary human islets were obtained from Prodo Labs. The cells were cultured in CMRL (Mediatech; 99-603-CV) + 10% FBS (GE Healthcare; SH3007003). After 24 h, cells were treated with BFA, TG, and CM (same as the stress treatment section). After 48 h, cells were collected, lysed in 500uL MPER buffer with protease complete, and incubated for 10 min shaking at 1500 RPM. Lysed cells were centrifuged for 10 min at 14,000 $g$ at 4 C and the supernatant was collected. 20ug of protein, as determined by BCA assay (Pierce) was mixed with 4x LDS buffer (Invitrogen), boiled for 5 min, and separated on a 4-12% Bis-Tris gel (Invitrogen) in MES running buffer (Invitrogen). The proteins were transferred onto PVDF membrane (BioRad), blocked in 5% milk (BioRad) for 1-h at room temperature, and incubated in primary antibodies in 5% milk overnight at 4 °C. The following day, blots were washed and incubated in secondary antibody for 1 h at room temperature. Blots were exposed with chemiluminescence (Biorad) and imaged on a Licor Odyssey FC. All blots were normalized to GAPDH. All antibody information and dilutions can be found in Supplementary Data 1.

## Statistics and reproducibility

No statistical method was used to predetermine the sample size. The scRNAseq is comprised of 3 separate patients sequenced using hashing in different batches. Two additional patients were sequenced using fixed scRNAseq. The ATAC-sequencing has 1 patient. Individual patient data can be found in Supplementary Data 1. Statistical significance of DEG was calculated through the Wilcoxon Rank Sum test with a fold change > 0.25 or < -0.25 and adjusted *p*-value using Bonferroni correction of $P < 0.05$, and differential motif accessibility was calculated through a logistic regression framework. For SC-islet in vitro experiments, we used unpaired or paired t-tests, Tukey's multiple comparisons, and Dunnets multiple comparisons to determine *P*-value. Significant values are represented as follows: *$P < 0.05$, **$P < 0.01$, ***$P < 0.001$, ****$P < 0.0001$, and not significant (ns) > 0.05.

**Reporting summary**

Further information on research design is available in the Nature Portfolio Reporting Summary linked to this article.

## Data availability

The single cell and single nuclei sequencing data generated in this study have been deposited in the gene expression omnibus (GEO) database and are accessible via accession code GSE237448. Source data are provided with this paper.

## Code availability

No codes were developed for the analysis of this study. All the sequencing data was analyzed using Seurat and Signac.

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

## Acknowledgements

This work was funded by the NIH (R01DK114233, R01DK127497, R01DK138469), JDRF (3-SRA-2023-1295-S-B), a Human Islet Research Network (HIRN) Catalyst Award, the Edward J. Mallinckrodt Foundation, and startup funds from the Washington University School of Medicine Department of Medicine to J.R.M. This work was partly supported by the

grants from the NIH (R01DK132090, R01DK020579) to F.Urano. This manuscript used data acquired from the Human Pancreas Analysis Program (HPAP-RRID: SCR_016202) Database (https://hpap.pmacs.upenn.edu), a Human Islet Research Network (RRID: SCR_014393) consortium (UC4DK112217, U01DK123594, UC4DK112232, and U01DK123716). The Washington University Institute of Clinical and Translational Sciences provided some support for sequencing and was supported by the NIH (UL1TR002345). Further support was provided by the Washington University Diabetes Research Center (P30DK020579). We thank the Genome Technology Access Center at the McDonnell Genome Institute at Washington University School of Medicine for help with genomic analysis. The Center is partially supported by NCI Cancer Center Support Grant P30CA91842 to the Siteman Cancer Center from the National Center for Research Resources (NCRR), a component of the National Institutes of Health (NIH), and NIH Roadmap for Medical Research. This publication is solely the responsibility of the authors and does not necessarily represent the official view of NCRR or NIH. M.M.M. was supported by the Cellular and Molecular Biology Training grant (T32GM139774). M.I. was supported by Rita Levi-Montalcini Postdoctoral Fellowship in Regenerative Medicine and the NIH (T32DK007120). D.A.V.-P. was supported by the NSF Graduate Research Fellowship Program (DGE-2139839 and DGE-1745038). L.V.-C. was supported by the NIH (F31DK125068). K.G.M. was supported by the NIH (T32DK108742). We would also like to thank Erika Brown (Washington University) for helpful feedback on the manuscript.

## Author contributions

M.M.M. and J.R.M. designed all the experiments and wrote the manuscript. M.M.M., P.A., M.I., K.G.M., J.S., and J.R.M. provided key contributions to the sequencing analysis. M.M.M., D.A.V-P., L.V-C., E.M., M.S., and S.E.G. contributed to the wet lab assays presented. F.Urano provided key guidance. All authors read, had the opportunity to provide feedback, and agreed to the manuscript.

## Competing interests

P.A., L.V.C., and J.R.M. are inventors of related patents and patent applications. D.A.V.P, L.V.C., and J.R.M. have stock in Sana Biotechnology. M.I. has stock in Vertex Pharmaceuticals. K.G.M. is currently employed by Vertex Pharmaceuticals. L.V.C. and J.R.M. were previously employed by Sana Biotechnology. F.Urano is an inventor of three patents related to the treatment of Wolfram syndrome and diabetes, US 9,891,231 "Soluble MANF in pancreatic beta cell disorders" and US 10,441,574 and US 10,695,324 "Treatment for Wolfram syndrome and other ER stress disorders." F.Urano is a Founder and President of CUR-E4WOLFRAM, INC., and Chair of the Scientific Advisory Board for Opris Biotechnologies and Emerald Biotherapeutics. F.Urano receives research funding from Prilenia, and Amylyx pharmaceuticals that are developing novel treatments for neurodegenerative disorders, optic nerve atrophy, diabetes, and Wolfram syndrome. The remaining authors declare no competing interests.
