## [Peer Review file · Nature Communications]

REVIEWER COMMENTS

Reviewer #1 (Remarks to the Author):

The manuscript by Maestas et al present a dataset of human islets treated with a number of stressors aimed to induce inflammatory/ER stress to pancreatic cell populations. The authors perform multiplex scRNA-seq following several treatments and perform multiple comparisons to identify gene lists corresponding to cell type- and stressor-specific signatures. Overall, this dataset is interesting and the validation of the work using stem-cell derived islets points to a novel gene as a modulator of islet response to stress. In particular, the multiplexing approach was successful and smart and can reduce some of the technical variability inherent to sequencing methodologies. However, several issues remain to be addressed before publication of this work:

Major issues:

- The authors should compare their CM treated dataset with the AAb+ and T1D donor datasets that have been made available through the HPAP consortium and recently presented here: PMID: 37188822 and <https://doi.org/10.1101/2023.02.03.526994>. In this way the information will become more disease relevant and the field can see how the in vitro treatment can correlate with the development of T1D. A similar approach was taken recently here for reference: PMID: 35941159 and the authors must cite and discuss the similarities and differences of the studies.

- Integration of human pancreas material has been proven challenging using a number of bioinformatics approaches. More details are needed in the integration process of the scRNA-seq data and also a detailed description of the code used is required.

- Indeed, the stressors chosen are commonly used in the literature for these purposes. However, prior to sequencing, these treatments should have been validated using quality control that the stressors work as intended (ER-stress induction/inflammation response) in the human islet system given the variability of human samples, activity of substances, experimental variations etc. How do the authors control for these variabilities and benchmark to published work in the literature?

- Is the effect of the cytokines reversible upon removal? Or do the phenotypes persist and if so in which cell types?

- Could you validate that the six commonly upregulated genes under all stressors are also upregulated on the protein level?

Minor issues:

- It is understandable that the source material is scarce but it would be nice to have more than 3 donors in the treatments since the islet preparations can have great variability in regards to gene expression. Also more than one donor for scATAC-seq experiment would be appreciated.
- In the scRNA-seq methods it is mentioned that pval is used for the differential expression statistics while in the statistics section it is mentioned that the FDR corrected is used. Since this manuscript is based largely on DEG analysis it really needs to be clarified which statistical measure was used as the FDR corrected one is the correct method used for this dataset.
- ER stress and activated UPR has been implicated in inducing proliferative responses. Is there such indication in the human islet system or in the CIB1 genetic knockdown/OE?
- Does the SC-islet composition change upon CIB1 knockdown?
- Is the drug rescue experiment possible to be performed on human islets additionally to SC-islets?
- These papers/preprints should be included in the discussion as they are relevant to this study:
PMID: 26099855, <https://doi.org/10.1101/2023.06.29.547000>,
<https://doi.org/10.1101/2023.02.16.528701>

Reviewer #2 (Remarks to the Author):

Nature Communications Maestas et al

Maestas and colleagues present scRNAseq data of human islets treated in vitro with 7 different stress inducers (plus the control group). The stressor include the well-known inducers such as a cytokine cocktail, brefeldin A and others. The authors perform the now standard analysis of UMAP clustering and differential gene expression analysis. They observe the expected induction of UPR response genes or inflammatory response signature depending on the stressor. Overall, in this

reviewer's opinion, the study does not reach the level of biological insight, novelty and impact normally associated with publications in Nature Communications

Specific comments:

The number of single cells analyzed per donor and treatment is actually quite small, and it is not clear from the analysis how reproducible the observed changes are from donor to donor. For example, for organ donor 1, there were a total of 3,450 cells, which split by 8 treatment groups comes to about 430 cells per treatment, of which again only small subset will be alpha cells or acinar ductal cells, for example. This puts into question the robustness of the data. Most of the plots appear to have treated the cells from the three organ donors in aggregate, making it impossible to assess reproducibility.

Some of the analyses, such as Figure 3, are reproducing the obvious (comparing endocrine to exocrine cells finds terms such as 'regulation of insulin secretion' and 'pancreas beta cells') or superficial (motif analysis in 3g includes motifs for TFs that are likely not expressed in the cell types analyzed).

The fold-changes in stress induced genes are not impressive, and again the values are not separated by donor, so not clear if they are reproducible.

Reviewer #3 (Remarks to the Author):

The manuscript by Maestas et. al. used single cell RNA-seq and single nuclei ATAC-seq to measure the transcriptomic changes in human cadaver islet under inflammatory (cytokine mix, CM) or ER stress (thapsigargin (TG) or BFA). The authors found numerous gene expression differences between control and stressed islet cells. ER stress gene signatures showed up in all stressed conditions including cytokine stress. To demonstrate the biological relevance of the single-cell analysis, the authors further used several inhibitors to show that suppressing the TG, BFA, or CM-induced signatures reduces apoptosis induced by BFA. The author further knock down CIB, a gene that is induced by all stressed conditions in all major endocrine cell types, and shows that the knockdown influences both GSIS and apoptosis. Lastly, the team used scATAC-seq to investigate the chromatin accessibility changes by BFA, TG, and CM. Overall the manuscript provides a useful resource for the islet research community. The transcriptomic changes reported are robust and consistent with several previous reports. The manuscript is well-written and the single-cell analysis

was carried out correctly. However, several issues need to be addressed before it is suitable for publication.

Major points:

1. The authors did a great job analyzing DEGs by cell types in the initial parts of the story. However, the focus soon turned to the common induced/suppressed gene across alpha, beta, and delta cells. And this eventually led to the validation experiment of KD CIB. While the commonly change genes indeed reflect the robust changes caused by a variety of stress, from the cellular perspective, the most fascinating phenomenon of islet dysfunction is the unique susceptibility of beta cells to stress. Only the beta cell undergoes severe damage and cell death with the same stress (ER stress or inflammation). This difference is likely the result of differentially regulated genes between beta and alpha/delta cells. Unfortunately, the authors discovered all the unique gene changes in beta cells but did not choose to precede exploring these genes or their upstream regulators.
2. The authors used a high dose of cytokine mix(1000ng/ml IFN-g, 100ng/ml IL1b, 500ng/ml TNFa). This is higher than many of the published studies. There is likely a widespread apoptosis over 48 hours at such a high level. Why choose such a harsh condition? Did the authors measure the basal level of cell death? At 48 hours, many of the transcriptomic responses are likely the secondary results. By ignoring the early responses and only looking at late responses, the design limited the mechanistic insights drawn from the study.
3. Figure 2A, the alpha cells have an equal or even more significant number of upregulated and downregulated genes. Why? Is this because the higher level of apoptosis in beta cells confounds the DEG analysis?
4. The authors showed multiple alpha and beta cell subpopulations in the initial analysis (Fig 1c). However, these subpopulations' definition and biological meaning is not clearly stated. For example, the beta cell population #1 contains mostly ctrl, TG, TNF, and IL1b but not the IFNg or IL1b+IFNg (Fig. S1d). Do the gene signatures in these subpopulations reveal any clues about the distinct events in the treatment?
5. Why do TG and BFA treatments have such a massive difference in DEG number? The TG is much weaker than BFA regarding gene changes in this study. But in fact, TG is a very potent ER stress inducer.
6. Figure 6H. I don't understand why it is showing the "Average accessibility" by every chromosome. This analysis doesn't make a lot of sense. If the authors' point is the widespread loss of accessibility, they should show the number of peaks or heat maps. The authors may want to suggest that the overall accessibility is changed because of the downregulation of several chromatin remodelers. However, the changes in INO80 are relatively minor (Fig 6g). Also, even in the case of knocking out chromatin remodelers, many studies, including in islet development, showed a specific pattern of chromatin accessibility loss. A more detailed analysis of these atac-seq data is needed to show the targets TF regulators of gain/loss peaks.

7. Knock-down of CIB led to a significant increase of basal insulin secretion (and corresponding reduced GSIS ratio). What is the mechanism of such a dramatic change? Does CIB directly control the insulin release?

Minor points:

1. Line 232, it should be “ABCC8”. Rather than “ABBC8”. The same typo showed up in the figure.
2. Line 250, the sentence is missing a few words and is not clear.
3. Line 303, “the list of binding partners.” Where is this list coming from? What binding partners? Interacting proteins or TF/co-factors that occupy the gene locus? This is not clearly stated.

REVIEWER COMMENTS

[Reviewers comments in black]

[Author comments in dark blue]

We thank the reviewer for the feedback.

[Reviewer]

Reviewer #1 (Remarks to the Author):

The manuscript by Maestas et al present a dataset of human islets treated with a number of stressors aimed to induce inflammatory/ER stress to pancreatic cell populations. The authors perform multiplex scRNA-seq following several treatments and perform multiple comparisons to identify gene lists corresponding to cell type- and stressor-specific signatures. Overall, this dataset is interesting and the validation of the work using stem-cell derived islets points to a novel gene as a modulator of islet response to stress. In particular, the multiplexing approach was successful and smart and can reduce some of the technical variability inherent to sequencing methodologies. However, several issues remain to be addressed before publication of this work:

Major issues:

- The authors should compare their CM treated dataset with the AAb+ and T1D donor datasets that have been made available through the HPAP consortium and recently presented here: PMID: 37188822 and <https://doi.org/10.1101/2023.02.03.526994>. In this way the information will become more disease relevant and the field can see how the in vitro treatment can correlate with the development of T1D.

[Authors]

We have added the single-cell sequencing data from the HPAP consortium and compared our CM dataset with T1D and AAB. We subsetted the beta cells from the AAB+, T1D, and non-diabetic donors from the HPAP consortium. We then conducted differential gene expression between AAB+ vs non-diabetic, T1D vs. non-diabetic, and CM vs. DMSO (our control). We then ran EnrichR on the upregulated differentially expressed genes to find pathways that were being differentially regulated between diabetic states and CM. This data can be found in Extended Data Fig. 12g, and in the text:

“We also compared our CM sequencing data with published T1D and AAB+ sequencing data⁵⁷. Using the upregulated DEG, we found pathways that were enrichment across all the different conditions. We found pathways related to antigen presentation and processing, cellular response to stress and stimuli, endosomal/vacuolar, and metabolism of proteins shared between T1D, AAB+, and CM (Extended Data Fig. 12g, Supplementary Table 16).”

In addition, we found other ways to relate our data to T1D:

“Interestingly, exocrine cells that have been treated with CM showed upregulation of genes associated with PD-1 (R-HSA-389948) and T1D (KEGG_2021) (Fig. 3c). Genes that

are upregulated are part of the major histone complex II (MHC Class II), including HLA-DRB5, HLA-DQA2, HLA-DQA1, and HLA-DPA1 (Fig. 3c, Supplementary Table 8, 9)."

"In patients with diabetes, studies have shown differential accessibility associated with T1D. Here, we investigate the risk variant rs7795896, which has previously been shown to have lower accessibility in T1D patients and is associated with lower CFTR expression³⁶. In our dataset, we find that the environmental factor of inflammation due to treatment with CM causes similar chromatin changes in ductal cells to that of a T1D patient (Fig. 3h)."

[Reviewer]

-A similar approach was taken recently here for reference: PMID: 35941159 and the authors must cite and discuss the similarities and differences of the studies.

[Authors]

We have added this important publication in several locations in the manuscript, now stating:

"Some of our findings are similar to that of a recent paper by Chen et al. investigating ER stress in β -cells³⁵. These authors found a β -cell specific signature comparing MIN6 cells to MEFs, in our paper we compared our primary human β -cells with α -, δ -, acinar, and ductal cells. To further investigate the β -cell-specific signature we conducted snATACseq and found only 5 genes that were upregulated under BFA stress and had enriched motif accessibility. In this study we were unable to explore these transcription factors, but these results could be used in future studies to better understand β -cell-specific stress responses. In addition, Chen et al. found loss of β -cell identity and the ability of the islet to recover from stress after recovery, which we saw in our model system as well. However, in this paper they also investigate translational differences, whereas in this paper we focus mainly on transcriptional differences compared between islet cell types."

"However, genes downregulated in endocrine tissue are related to the integration of energy metabolism (Reactome_2022), circadian entrainment (KEGG_2021), and genes associated with lysosomes, which has previously been shown to be downregulated during ER stress³⁵."

"We treated cadaveric human islets with CM for 48-hr and then removed and washed off the CM and found a reduction of MT2A, CXCL11, CXCL9, IL32, SOD2, ISG15, SAA2, and LCN2 when compared to CM treated cells (Extended Data Fig. 6b). This recovery from treatment has been shown in other systems as well³⁵."

"Previous studies have investigated cytokines^{24,27,35,42,68-70} and ER stress^{15,71} in the context of T1D utilizing cell lines, other model animals, or bulk sequencing. Our use of single-cell/nuclei sequencing offers additional in-depth analysis of cell-type-specific changes associated with ER and inflammatory stress."

"A recent study using ribosomal footprinting, identified translational changes during ER stress³⁵ and future analysis of lipidomic, metabolomics, and other post-translational

modifications could add to our understanding of ER and inflammatory stress aiding diabetes development¹¹²⁻¹¹⁵.”

[Reviewer]

- Integration of human pancreas material has been proven challenging using a number of bioinformatics approaches. More details are needed in the integration process of the scRNA-seq data and also a detailed description of the code used is required.

[Authors]

We have added a more detailed description under methods, now saying:

“Then the three hashed patients were individually LogNormalized using Normalize Data and the top 2000 variable features were identified using FindVariableFeatures. FindIntegrationAnchors function was used to identify anchors between the three patients. The anchors will then be used to integrate the patients using IntegrateData. Following integration, ScaleDate, RunPCA, and RunUMAP were used on the integrated data set containing data from the three patients. Clustering and dimension reduction were done using dimensions of 1:20 and a resolution of 0.3. Pair-wise comparisons using the function FindMarkers, and the Wilcoxon Rank Sum test was used to determined DEG with a log fold change of at least 0.25 and adjusted pvalue<0.05 using Bonferroni correction. EnrichR was used for all gene set enrichment analyses with a p value <0.05.”

[Reviewer]

- Indeed, the stressors chosen are commonly used in the literature for these purposes. However, prior to sequencing, these treatments should have been validated using quality control that the stressors work as intended (ER-stress induction/inflammation response) in the human islet system given the variability of human samples, activity of substances, experimental variations etc. How do the authors control for these variabilities and benchmark to published work in the literature?

[Authors]

We validated our treatments using markers from published literature as a quality control. To clarify this, we have added data showing the z-score of 5 genes found in the literature that are upregulated under BFA, TG, or CM. in Extended Data Fig. 3f. Furthermore, in the text we now say:

“To validate our stressors, we assessed the expression of genes from published literature^{15,28,29} and found upregulation of HERPUD1, HSPA5, COPZ2, GOLGA2, GBPF, CREB3, and GOG2 under BFA stress; MANF, SDF2L1, PPP1R15A, FKBP11, DDIT3, ATF4, and HERPUD1 under TG stress; and STAT1, CXCL10, NOS2, RSAD2, ISG20, CD40, and OAS1 under CM stress (Fig. 1e, Extended Data Fig.3f). This data was consistent across all five patients (Extended Data Fig. 3f). We validated our control cell population through comparative analysis with a published dataset of cadaveric human islets from donors without diabetes³⁰ (Extended Data Fig. 1j-k). We also conducted a Pearson correlation between hashed DMSO and fixed DMSO samples and found consistency across cell types (Extended Data Fig. 3g-h).”

[Reviewer]

- Is the effect of the cytokines reversible upon removal? Or do the phenotypes persist and if so in which cell types?

[Authors]

We performed this new experiment and indeed confirmed that most of the effects of the cytokines are reversed upon removal. We now have this data in Extended Data Fig 6b and now say:

“We also wanted to determine if islets can recover from these inflammatory conditions. We treated cadaveric human islets with CM for 48-hr and then removed and washed off the CM and found a reduction of MT2A, CXCL11, CXCL9, IL32, SOD2, ISG15, SAA2, and LCN2 when compared to CM treated cells (Extended Data Fig. 6b). This recovery from treatment has been shown in other systems as well³⁵. MAFA was reduced in islets treated with CM however after recovery, the cells were able to induce MAFA expression. This data indicates that human islets can recover after cytokine stress even with significant transcriptional changes during inflammation.”

[Reviewer]

- Could you validate that the six commonly upregulated genes under all stressors are also upregulated on the protein level?

[Authors]

We conducted this proposed experiment using western blots. This data is now in Extended Data Fig 9d, and in the text we now say:

“Protein expression of these genes varies across stressors, which could be due to looking at whole islet levels and not at a single-cell resolution (Extended Data Fig. 9d). However, HSP90B1 is upregulated under BFA and TG stress.”

[Reviewer]

Minor issues:

- It is understandable that the source material is scarce but it would be nice to have more than 3 donors in the treatments since the islet preparations can have great variability in regards to gene expression. Also more than one donor for scATAC-seq experiment would be appreciated.

[Authors]

We have increased the number of human islet donors to a total of five, now representing a total of 111,648 cells. This new data can be seen in Extended Data Fig 2 and 3. As the reviewer anticipated, sourcing this material has been difficult due to its scarcity, made even more complicated due to the complex downstream analyses of our various workflows in this study. As a result of islet scarcity, even after getting an extension from the editor to give us more time to secure additional islets, we were not able to get islets for all new desired experiments for this revision and had to prioritize the experiments we used these islets for. Since the sn-ATACseq data was a small component in the overall

claims of this study and this issue was also listed as minor, we were not able to add further information. Instead, in the text, we now address this limitation by saying:

“However, a limitation of this study is that we were only able to secure one islet donor for the sn-ATAC data, and therefore cannot speak as to how common these findings are across donors. This study could be expanded upon by including a larger number of donors better representing the population, material from patients with diabetes, and a wider range of ages⁷⁴⁻⁷⁷.”

[Reviewer]

- In the scRNA-seq methods it is mentioned that pval is used for the differential expression statistics while in the statistics section it is mentioned that the FDR corrected is used. Since this manuscript is based largely on DEG analysis it really needs to be clarified which statistical measure was used as the FDR corrected one is the correct method used for this dataset.

[Authors]

We used the FDR correct P-value (adjusted p-value) and have clarified this in the manuscript in the methods section “single-cell RNA sequencing analysis”, now saying:

“Pair-wise comparisons using the function FindMarkers, and the Wilcoxon Rank Sum test was used to determined DEG with a log fold change of at least 0.25 and adjusted pvalue<0.05 using Bonferroni correction.”

[Reviewer]

- ER stress and activated UPR has been implicated in inducing proliferative responses. Is there such indication in the human islet system or in the CIB1 genetic knockdown/OE?

[Authors]

We have added new data to investigate this idea. We found proliferation/cell cycle markers in our single-cell data and found upregulation under BFA stress. This data is now in Fig. 6c and Extended Data Fig. 12a and now say in the text:

“We also found that BFA induces upregulation of proliferation and cell cycle markers⁵⁶ in β - and α -cells, however, MKI67 had the same expression across all conditions (Fig. 6c, Extended Data Fig. 12a).”

We also added new data determining the proliferation in SC-islets when CIB1 was knocked down or overexpressed using Ki-67 immunostaining in Extended Data Fig. 10h-i. We found decreases in proliferation after CIB1 overexpression or KD under TG stress. In the text, we now say:

“Next, we assessed the proliferation of CIB1 KD or OE under stress conditions. We found that both KD and OE induced reduction of Ki-67+ cells under TG stress. We see no significant difference between control and CIB1 KD in other stress conditions (Extended Data Fig. 10h-i).”

[Reviewer]

- Does the SC-islet composition change upon CIB1 knockdown?

[Authors]

We have added this new data (Extended Data Fig. 10g) and found that there is no significant difference in SC-islet composition. In the text, we now say:

“We also find that CIB1 KD or CIB1 OE did not significantly change cell type proportions (Extended Data Fig. 10g).”

[Reviewer]

- Is the drug rescue experiment possible to be performed on human islets additionally to SC-islets?

[Authors]

As the reviewer anticipated, sourcing this material has been difficult due to its scarcity, made even more complicated due to the complex downstream analyses of our various workflows in this study. As a result of islet scarcity, even after getting an extension from the editor to give us more time to secure additional islets, we were not able to get islets for all new desired experiments for this revision and had to prioritize the experiments we used these islets for. Since the drug rescue experimental data was a small component in the overall claims of this study and this issue was also listed as minor, we were not able to add further information. Instead, in the text, we now address this limitation by saying:

“By regulating gene expression through genetic editing or the addition of a drug, we were able to reduce apoptosis in SC-islets, establishing the potential to improve islet health. A limitation of our study is that we were not able to source primary human islets for all components of our study.”

[Reviewer]

- These papers/preprints should be included in the discussion as they are relevant to this study: PMID:26099855, <https://doi.org/10.1101/2023.06.29.547000>, <https://doi.org/10.1101/2023.02.16.528701>

[Authors]

Thank you for pointing these manuscripts out. We have now cited them and included relevant discussion, now saying:

“Previous studies have investigated cytokines^{24,27,35,42,68–70} and ER stress^{15,71} in the context of T1D utilizing cell lines, other model animals, or bulk sequencing. Our use of single-cell/nuclei sequencing offers additional in-depth analysis of cell-type-specific changes associated with ER and inflammatory stress.”

“IFN γ induced the greatest transcriptional changes in β -cells compared to the other individual cytokines, which has previously been reported³⁹.”

Reviewer #2 (Remarks to the Author):

[Reviewers comments in black]

[Author comments in dark blue]

We thank the reviewer for the feedback.

[Reviewer]

Nature Communications Maestas et al

Maestas and colleagues present scRNAseq data of human islets treated in vitro with 7 different stress inducers (plus the control group). The stressor include the well-known inducers such as a cytokine cocktail, brefeldin A and others. The authors perform the now standard analysis of UMAP clustering and differential gene expression analysis. They observe the expected induction of UPR response genes or inflammatory response signature depending on the stressor. Overall, in this reviewer's opinion, the study does not reach the level of biological insight, novelty and impact normally associated with publications in Nature Communciations

We have clarified the text to better communicate to the readers the novelty and importance of this new study. In the text, we now say:

“Previous studies have investigated cytokines^{24,27,35,42,69–71} and ER stress^{15,72} in the context of T1D utilizing cell lines, other model animals, or bulk sequencing. Our use of single-cell/nuclei sequencing offers additional in-depth analysis of cell-type-specific changes associated with ER and inflammatory stress.”

“We wanted to establish the utility of this dataset for drug discovery. Using The Drug Gene Interaction Database (DGIdb)⁴⁷ and DrugBank we explored drug-gene interactions. We found three candidate genes RELA (subunit NF-KB), CKB (enzyme to transfer phosphates), and P4HB (part of oxidoreductase complex), which were all upregulated under BFA stress in α - and β -cells (Fig. 4a). The corresponding drugs are Cynaropicrin, an inhibitor of RELA⁴⁸, Acyclovir, which is an antiviral agent previously shown to help insulin resistance in a mouse model⁴⁹, and Liquiritin, which has shown anti-inflammatory effects⁵⁰.”

“We found all three drugs significantly reduced apoptosis in SC-islets under BFA stress, and Cynaropicrin has a dose-dependent reduction of apoptosis (Fig. 4d). These data reveal how the scRNAseq data set can be leveraged to discover approaches to improve SC-islet health.”

“Here, we discovered that CIB1 is important for the function of SC-islets and response to apoptosis. Islet identity and UPR gene expression were affected under both homeostatic and stress conditions when CIB1 was KD or OE.”

“Previously, T1D risk variant rs7795896 at the CFTR locus was identified to have lower accessibility³⁶, and here we found that CM stress also decreased accessibility in this region within exocrine tissue. Taken together, this suggests that both genetics and microenvironmental cues, like stress, could play a role in developing T1D via rs7795896.”

“To further investigate the β -cell-specific signature we conducted snATACseq and found only 5 genes that were upregulated under BFA stress and had enriched motif accessibility.”

“To determine if the gene expression of these associated TF-binding motifs also increases, we compared the enriched motifs with upregulated genes under BFA stress and found 44 genes/motifs upregulated (Fig. 6k). Of these 44, only 5 were unique to β -cells (Fig. 6k, Supplementary Table 17): BACH2, a risk gene for T1D⁶⁴; SMAD5, a protein downstream of TGF- β signaling and can autonomously promote glycolysis⁶⁵; HMBOX1, a regulator of telomerase activity⁶⁶; TP53, a gene connected T1D and T2D by TP53-mediated apoptosis⁶⁷; and JUND, a gene associated with β -cell dysfunction⁶⁸.”

“We analyzed different UPR pathways and ERAD across cell types and stressors and found large cell-type-specific responses of these pathways to specific stressors (Fig. 2b)... Taken together, our analysis reveals cell-type-specific transcriptional changes and stress response mechanisms, highlighting the distinct molecular signatures and differential regulation of the UPR and ERAD pathways in various endocrine and exocrine cell types under different stress conditions.”

This study is the first to use single-cell sequencing to perform in-depth analysis of human islets under a wide range of stressors associated with diabetes. These stressors were chosen because of their use and acceptance in studying islet stress. One immediate impact of this work is that it provides the field with cell-type-specific stress signatures of individual cell types in the islet and pancreas. In other words, we have the transcriptional and chromatin accessibility of each individual cell type in response to defined diabetes-associated stress. While we indeed confirmed that already known UPR or other stress response signatures were present, we also used these datasets to make many novel and important findings. We list out the common and unique transcriptional and epigenetic responses to each stress across each cell type, such as alpha cells, beta cells, delta cells, ductal cells, and acinar cells. There were many unexpected findings described throughout the study, such as cell type specific differences in UPR pathways and ERAD to specific stressors. Some other additional discoveries include that inflammatory cytokines regulate the chromatin accessibility for the T1D risk variant rs7795896 around *CFTR*, *CIB1* regulates insulin secretion and apoptosis under stress, and Cynaropicrin, Acyclovir, and Liquiritin help protect cells from stress. The hope is that these significant and new findings will help to advance the field by leading to the discovery of approaches to slow down loss of the remaining islet cell mass or improve survival of transplanted islets in patients with diabetes.

Specific comments:

The number of single cells analyzed per donor and treatment is actually quite small, and it is not clear from the analysis how reproducible the observed changes are from donor to donor. For example, for organ donor 1, there were a total of 3,450 cells, which split by 8 treatment groups comes to about 430 cells per treatment, of which again only small subset will be alpha cells or acinar ductal cells, for example. This puts into question the robustness of the data. Most of the plots appear to have treated the cells from the three organ donors in aggregate, making it impossible to assess reproducibility.

[Authors]

We have added new data from additional donors to address this concern. This brings us to a total of 111,648 cells sequenced, including these cell numbers across treatment groups: BFA =7,554; CM=11,528; CTRL=13,163; IFNG=15,345; IL1B=15,995; IL1B+IFNG=18,671; TG=14,172; TNFA=15,220. Analysis of these new datasets can be found in Extended Data Fig. 2 and 3. We have also separated all 5 donors and compared the expression of genes across stressors in Extended Data Fig 1, 2, and 3. Overall, we found the key findings of this study to be consistent even with the increase in the number of cells and donors considered, representing the robustness of this novel dataset we are providing the scientific community.

To better address the reproducibility of this data set we have added additional text and data to the manuscript including:

“The treated cadaveric human islets were sequenced and we ensured reproducibility by assessing each patient data individually. We found across all five patients, top differentially expressed genes had similar expression levels (Extended Data Fig. 1d-f, 2c-d, Supplementary Table 2-6). In addition, across all five patients we observed similar proportions of cells across all conditions, cell type populations, and sequencing methods (Extended Data Fig. 3a-e).”

“To validate our stressors, we assessed the expression of genes from published literature^{15,28,29} and found upregulation of HERPUD1, HSPA5, COPZ2, GOLGA2, GBPF, CREB3, and GOG2 under BFA stress; MANF, SDF2L1, PPP1R15A, FKBP11, DDIT3, ATF4, and HERPUD1 under TG stress; and STAT1, CXCL10, NOS2, RSAD2, ISG20, CD40, and OAS1 under CM stress (Fig. 1e, Extended Data Fig.3f). This data was consistent across all five patients (Extended Data Fig. 3f). We validated our control cell population through comparative analysis with a published dataset of cadaveric human islets from donors without diabetes³⁰ (Extended Data Fig. 1j-k). We also conducted a Pearson correlation between hashed DMSO and fixed DMSO samples and found consistency across cell types (Extended Data Fig. 3g-h).”

“We also conducted a Pearson correlation between hashed DMSO and fixed DMSO and found consistent cell types and high correlations (Extended Data Fig. 3g-h).”

[Reviewer]

Some of the analyses, such as Figure 3, are reproducing the obvious (comparing endocrine to exocrine cells finds terms such as ‘regulation of insulin secretion’ and ‘pancreas beta cells’) or superficial (motif analysis in 3g includes motifs for TFs that are likely not expressed in the cell types analyzed).

[Authors]

The majority of the gene set analysis in Figure 3 relate to non-obvious or non-superficial biology, such as metabolism, immunology, and various signaling pathways. We originally included a small number of gene sets that relate more specifically exocrine or endocrine pancreatic biology to provide evidence of robustness, showing we see expected changes in specific tissue types and also to pre-emptively address the potential for transdifferentiation.

However, given the concerns raised by this reviewer and the potential to confuse other readers of this manuscript, we have removed the listed gene sets of concern from Figure 3 and instead have all these gene sets listed in Supplemental Table 3.

[Reviewer]

The fold-changes in stress induced genes are not impressive, and again the values are not separated by donor, so not clear if they are reproducible.

[Authors]

We have clarified this in the manuscript:

“Pair-wise comparisons using the function FindMarkers, and the Wilcoxon Rank Sum test was used to determined DEG with a log fold change absolute value of $> |0.25|$ and adjusted p -value <0.05 using Bonferroni correction. This threshold was set based on previously published data^{27,32,39,65,92,93,101}.”

To further clarify, in a recent paper by Fasolino et al. in nature metabolism, the authors compared different cell types in control, AAb+, and T1D and found Log_2FC at the highest to be from about -4.5 to 4.5, with most between 1-3 absolute log_2FC . In our data set, the Cytokine treatment induces a Log_2FC of about -4 to 7. While the BFA treatment induces about -6 to 4, most of our genes also are around 1-3 absolute log_2FC . In addition, these authors use a Log_2FC threshold of >0.1 , we are more stringent in our paper using $\text{Log}_2\text{FC}>0.25$ for DEGs. In another paper by Colli M et al, in nature communications, the authors compared primary islets treated with/without IFNa and found at 2-hr a Log_2FC between -5 and 7.8, at 8-hr between -4.5 and 14, and at 24 hr between -3.5 and 8.6. This shows a reduction of fold-changes after time, in our study we look at 48-hr which could be inducing fold-change differences. In this paper they used a Log_2FC threshold of $> |0.58|$.

In Figure 4, these Log_2FC are lower because the highly differentially expressed genes are not cell type specific and are upregulated in many different cell types, most of these genes are related to ER stress or inflammatory genes (depending on the stressor) (Supplementary Table 10)

Now in Extended Data Fig 1-3 we have compared across all 5 donors separating by donor. In panels 1d-f and 2c-d, all these genes are differentially expressed between treatment and control, and across all patients the genes have similar expression profiles. In Extended Data Fig. 3e we now show the expression level of key cell identity genes across donors. Extended Data Fig. 3f shows key genes from published literature validating stressors are responding as expected, this is shown across patients as well. In Extended Data Fig. 3h we conducted a Pearson correlation to compare our control hashed samples with our fixed samples and found great correlation across cell types. The individual patient differentially expressed genes can now be found in supplemental Data Tables 2-6.

We have added text to the paper to include:

“The treated cadaveric human islets were sequenced, and we ensured reproducibility by assessing each patient data individually. We found across all five patients, top differentially expressed genes had similar expression levels (Extended Data Fig. 1d-f, 2c-

d, Supplementary Table 2-6). In addition, across all five patients we observed similar proportions of cells across all conditions, cell type populations, and sequencing methods (Extended Data Fig. 3a-e)."

"Across patients, we observed similar proportions of cells across all conditions, cell type populations, and sequencing methods (Extended Data Fig. 3a-e). "

"To validate our stressors, we assessed the expression of genes from published literature^{15,28,29} and found upregulation of HERPUD1, HSPA5, COPZ2, GOLGA2, GBPF, CREB3, and GOG2 under BFA stress; MANF, SDF2L1, PPP1R15A, FKBP11, DDIT3, ATF4, and HERPUD1 under TG stress; and STAT1, CXCL10, NOS2, RSAD2, ISG20, CD40, and OAS1 under CM stress (Fig. 1e, Extended Data Fig.3f). This data was consistent across all five patients (Extended Data Fig. 3f). We validated our control cell population through comparative analysis with a published dataset of cadaveric human islets from donors without diabetes³⁰ (Extended Data Fig. 1j-k). We also conducted a Pearson correlation between hashed DMSO and fixed DMSO samples and found consistency across cell types (Extended Data Fig. 3g-h)."

"We also conducted a Pearson correlation between hashed DMSO and fixed DMSO and found consistent cell types and high correlations (Extended Data Fig. 3g-h)."

Reviewer #3 (Remarks to the Author):

[Reviewers comments in black]

[Author comments in dark blue]

We thank the reviewer for the feedback.

[Reviewer]

The manuscript by Maestas et. al. used single cell RNA-seq and single nuclei ATAC-seq to measure the transcriptomic changes in human cadaver islet under inflammatory (cytokine mix, CM) or ER stress (thapsigargin (TG) or BFA). The authors found numerous gene expression differences between control and stressed islet cells. ER stress gene signatures showed up in all stressed conditions including cytokine stress. To demonstrate the biological relevance of the single-cell analysis, the authors further used several inhibitors to show that suppressing the TG, BFA, or CM-induced signatures reduces apoptosis induced by BFA. The author further knock down CIB, a gene that is induced by all stressed conditions in all major endocrine cell types, and shows that the knockdown influences both GSIS and apoptosis. Lastly, the team used scATAC-seq to investigate the chromatin accessibility changes by BFA, TG, and CM. Overall the manuscript provides a useful resource for the islet research community. The transcriptomic changes reported are robust and consistent with several previous reports. The manuscript is well-written and the single-cell analysis was carried out correctly. However, several issues need to be addressed before it is suitable for publication.

Major points:

1. The authors did a great job analyzing DEGs by cell types in the initial parts of the story. However, the focus soon turned to the common induced/suppressed gene across alpha, beta, and delta cells. And this eventually led to the validation experiment of KD CIB. While the commonly change genes indeed reflect the robust changes caused by a variety of stress, from the cellular perspective, the most fascinating phenomenon of islet dysfunction is the unique susceptibility of beta cells to stress. Only the beta cell undergoes severe damage and cell death with the same stress (ER stress or inflammation). This difference is likely the result of differentially regulated genes between beta and alpha/delta cells. Unfortunately, the authors discovered all the unique gene changes in beta cells but did not choose to precede exploring these genes or their upstream regulators.

[Authors]

Thank you for this comment, we looked further to understand if *CIB1* KD affected other cell types within the SC-islet. We looked at the expression of key hormone genes and found that *CIB1* KD is only affecting C-peptide protein expression, this data can be found in extended data fig. 10f and as below:

“Flow cytometry confirmed a decrease in C-Peptide expression in CIB1 KD cells and no significant difference in CIB1 OE (Extended Data Fig. 10f). In addition, even though we saw upregulation of CIB1 in our sequencing data in all three cell types and differences in GCG expression after CIB1 KD or OE in SC-islets (Extended Data Fig. 11a), we see no significant differences in GCG or SST protein expression between CIB1 KD or CIB1 OE compared to control SC-islets (Extended Data Fig. 10f). We also find that CIB1 KD or CIB1 OE did not significantly change cell type proportions (Extended Data Fig. 10g).”

We also investigated protein expression of beta cell-specific targets from Fig 4b:

“We validated IDI1, upregulated under BFA stress, and RAB3B, upregulated under TG stress using immunofluorescence (Extended Data Fig. 8a). RRAD, which is upregulated under CM stress showed no obvious differences in protein levels between treatments.”

Here we were unable to explore all the possible beta cell-specific targets but have added a statement in the discussion to address this:

“To further investigate the β -cell-specific signature we conducted snATACseq and found only 5 genes that were upregulated under BFA stress and had enriched motif accessibility. In this study we were unable to explore these transcription factors, but these results could be used in future studies to better understand β -cell-specific stress responses.”

We have also added additional data to our β -cell-specific figure that could be a resource for further studies:

“We used JASPER2020 to identify motif accessibility across β -, α -, and δ -cells (Fig. 6i, Supplementary Table 17). We compared the enriched motifs with upregulated genes under BFA stress and found 44 genes/motifs upregulated (Fig. 6k). Of these 44, only 5 were unique to β -cells (Fig. 6k, Supplementary Table 17): BACH2, a risk gene for T1D⁶³; SMAD5, a protein downstream of TGF- β signaling and can autonomously promote glycolysis⁶⁴; HMBOX1, a regulator of telomerase activity⁶⁵; TP53, a gene connected T1D and T2D by TP53-mediated apoptosis⁶⁶; and JUND, a gene associated with β -cell dysfunction⁶⁷.”

“We also compared RNA expression and promoter accessibility of genes that were significantly upregulated in β -cells but not α - or δ -cells (Fig. 6k). These genes included WSB1, a gene involved in hypoxia⁶¹, TNFRSF10, shown in other models to induce downstream inflammatory cytokine production after ER stress⁶², and SMARCE1, a chromatin remodeler.”

[Reviewer]

2. The authors used a high dose of cytokine mix(1000ng/ml IFN-g, 100ng/ml IL1b, 500ng/ml TNFa). This is higher than many of the published studies. There is likely a widespread apoptosis over 48 hours at such a high level. Why choose such a harsh condition? Did the authors measure the basal level of cell death? At 48 hours, many of the transcriptomic responses are likely the secondary results. By ignoring the early responses and only looking at late responses, the design limited the mechanistic insights drawn from the study.

[Authors]

We have added data demonstrating that this concentration of cytokine mix (CM) does not affect the viability of primary islets or SC-islets in Extended Data Figure 6a and 9a. In the main text, we have clarified this and limitations of the study, now saying:

“We found upregulation of apoptotic-associated pathways in our gene ontology analysis, prompting us to investigate if our CM cocktail was inducing cell death. We found no

significant difference in the viability of cadaveric human islets when treated with CM compared to control (Extended Data Fig. 6a)."

"In this study, we focus on 48-hr treatments of human islets, while acute responses by reducing treatment timing or investigating chronic responses using long-term culture microphysiological systems¹⁰⁴⁻¹⁰⁶ would also be of interest to investigate in the future."

Also of note, there is substantial variation in the literature in terms of concentration and timing, including comparably high treatments in prior studies in Nature Communications (PMID: 27163171) and Science Translational Medicine (PMID: 32321868).

[Reviewer]

3. Figure 2A, the alpha cells have an equal or even more significant number of upregulated and downregulated genes. Why? Is this because the higher level of apoptosis in beta cells confounds the DEG analysis?

[Authors]

We have added data from two additional donors, now having 31,603 beta cells represented in the single-cell sequencing data. Furthermore, additional data in Extended Data Figure 6a doesn't show a decrease in viable cell counts with stress treatment.

Thank you for this comment we also found this result interesting, however, this is a long-standing issue within the field about why the beta cells are preferentially attacked in type 1 diabetes. This study is a step toward better understanding the transcriptional responses of each cell type to the stress response underlying diabetes.

Interestingly, a paper published in 2018 by Brissova et al. indicated differences in alpha cell transcription during T1D. In this study, they investigated islets from T1D donors and the islets had reduced glucagon secretion, even with having more alpha cells present. In addition, they found a reduction of alpha cell identity markers and increases in UPR genes. This was done using bulk RNA-seq so they were not able to compare alpha and beta cells.

We have now added this to our manuscript to address this concern:

"During the progression of T1D, β -cells are preferentially attacked by the immune system and are killed⁴⁴, while the other major endocrine cell type, α -cells, have reduced glucagon secretion and gene expression⁵⁵. The reason for the different effects on islet cell types during diabetes is unknown. "

[Reviewer]

4. The authors showed multiple alpha and beta cell subpopulations in the initial analysis (Fig 1c). However, these subpopulations' definition and biological meaning is not clearly stated. For example, the beta cell population #1 contains mostly ctrl, TG, TNF, and IL1b but not the IFNg or IL1b+IFNg (Fig. S1d). Do the gene signatures in these subpopulations reveal any clues about the distinct events in the treatment

[Authors]

We have clarified this in the text by now stating:

“These sub-populations are not donor-specific and, instead, arise from the different exogenous stressors (Extended Data Fig. 1g-h). α -cell 1 population is composed of control, TG, and IL1 β treated cells and expressed high levels of α -cell markers (Fig. 1b-d, Extended Data Fig. 1i). α -cell 2 was treated with CM, IL1 β +IFN γ , IL1 β , and TNF α and showed increase expression of inflammatory markers. α -cell 3 population had only IFN γ treated cells and showed upregulation of inflammatory and α -cell markers. 85.1% of α -cell population 4 were treated with BFA and we saw a decreased expression of α -cell markers and upregulation of UPR-associated genes (Extended Data Fig. 1i, Supplementary Table 7). β -cells show a similar sub-clustering of stressors where β -cell population 1 was treated with control, TG, TNF α , and IL1 β and had the highest expression of β -cell markers, while population 2 was treated with CM, IL1 β +IFN γ , and IFN γ inducing increased inflammatory gene expression (Extended Data Fig. 1i). 94.6% of the cells in β -cell population 3 were treated with BFA and have decreased β -cell markers compared to other β -cell populations (Fig. 1d, Extended Data Fig. 1i, Supplementary Table 7).”

[Reviewer]

5. Why do TG and BFA treatments have such a massive difference in DEG number? The TG is much weaker than BFA regarding gene changes in this study. But in fact, TG is a very potent ER stress inducer.

[Authors]

TG and BFA are chemicals with very different biochemistries, so we would expect differential responses of cells treated with these compounds. While TG is a potent ER stress inducer, BFA targets vesicular transport from the ER to the Golgi causing both ER and Golgi stress. In other words, it is expected that BFA would have a greater effect on cells given its expected biochemical responses. We have clarified this in the text by now saying:

“BFA stress, which affects both the Golgi and ER³², resulted in a robust transcriptional response with the most upregulated genes in β - (2,679 genes), α - (2,616 genes), and ductal cells (1,399 genes).”

[Reviewer]

6. Figure 6H. I don't understand why it is showing the “Average accessibility” by every chromosome. This analysis doesn't make a lot of sense. If the authors' point is the widespread loss of accessibility, they should show the number of peaks or heat maps. The authors may want to suggest that the overall accessibility is changed because of the downregulation of several chromatin remodelers. However, the changes in INO80 are relatively minor (Fig 6g). Also, even in the case of knocking out chromatin remodelers, many studies, including in islet development, showed a specific pattern of chromatin accessibility loss. A more detailed analysis of these atac-seq data is needed to show the targets TF regulators of gain/loss peaks.

[Authors]

We have clarified this by now showing the number of differentially accessible regions (DAR) between DMSO and BFA in beta cells instead of average accessibility this data can be found in Fig. 6j. In the text we now state:

“We next assessed differentially accessible regions (DAR) between BFA and DMSO in β -cells and found 8294 regions significantly (Adjusted P-value <0.05) differentially accessible (Fig. 6i, Supplementary Table 17) and 3819 regions that had an absolute average Log₂Fold change of at least 0.1. DAR associated with INS was significantly closed under BFA stress, while RABEPK, associated with endosomal trans-golgi network transport⁶⁰ was more open under BFA stress compared to DMSO (Fig. 6j). In α -cells, 4979 regions were significantly differentially accessible, with 2,761 regions with an absolute average Log₂Fold change of at least 0.1 (Extended Data Fig. 13d). In δ -cells, there were only 1216 DAR and 1,043 met our threshold of 0.1 Log₂Fold (Extended Data Fig. 13e).”

We have also made heatmaps for the chromatin remodelers to better compare across stress conditions in β -cells. This can be found in Fig. 6h and in the text saying:

“We were next interested in understanding if chromatin remodelers⁵⁸ are affected during ER stress in β -cells. Other model systems show recruitment of remodelers after induction of ER stress⁵⁹. Here, we found upregulation of genes associated with chromatin remodelers mainly under BFA-induced stress in β -cells (Fig. 6h).”

To further look at transcription factors (TF) we investigated TF-binding motif accessibility in Fig. 6i. We have added to the manuscript saying:

“We used JASPER2020 to identify motif accessibility across β -, α -, and δ -cells (Fig. 6i, Supplementary Table 17). Motif enrichment in β -cells shows an increase in β -cell identity, such as NKX6.1, SIX2, and ONECUT2⁶³. To determine if the gene expression of these associated TF-binding motifs also increases, we compared the enriched motifs with upregulated genes under BFA stress and found 44 genes/motifs upregulated (Fig. 6k). Of these 44, only 5 were unique to β -cells (Fig. 6k, Supplementary Table 17): BACH2, a risk gene for T1D⁶⁴; SMAD5, a protein downstream of TGF- β signaling and can autonomously promote glycolysis⁶⁵; HMBOX1, a regulator of telomerase activity⁶⁶; TP53, a gene connected T1D and T2D by TP53-mediated apoptosis⁶⁷; and JUND, a gene associated with β -cell dysfunction⁶⁸.”

In addition, we added data comparing upregulated genes under BFA stress to the accessibility of the associated promoter:

“We also compared RNA expression and promoter accessibility of genes that were significantly upregulated in β -cells but not α - or δ -cells (Fig. 6k). These genes included WSB1, a gene involved in hypoxia⁶¹, TNFRSF10, shown in other models to induce downstream inflammatory cytokine production after ER stress⁶², and SMARCE1, a chromatin remodeler.”

[Reviewer]

7. Knock-down of CIB led to a significant increase of basal insulin secretion (and corresponding reduced GSIS ratio). What is the mechanism of such a dramatic change? Does CIB directly control the insulin release?

[Authors]

We have clarified about our understanding of CIB1 in the manuscript, now saying:

“CIB1 has many roles in the cell, one being a protein-ligand of inositol-1,4,5-triphosphate receptor (InsP3R)^{53,90} and can then inhibit InsP3 binding, regulating intracellular calcium. When we KD CIB1 this might be allowing for the release of calcium from the ER resulting in increased cytosolic calcium and subsequent secretion of stored insulin under low glucose conditions. This mechanism would need to be investigated further in future studies.”

In other words, we believe that the relationship of CIB1 with insulin secretion primarily relates with its relationship with cytosolic calcium. Reduction of CIB1 seems to increase cytosolic calcium, resulting in increased basal insulin secretion and reduced relative glucose responsive secretion.

[Reviewer]

Minor points:

1. Line 232, it should be “ABCC8”. Rather than “ABBC8”. The same typo showed up in the figure.

[Authors]

This has been fixed in the manuscript to reflect the correct gene name.

[Reviewer]

2. Line 250, the sentence is missing a few words and is not clear.

[Authors]

We have corrected the wording to now say:

“In α -cells, CXCL6 (chemokine), GC (binds vitamin D), ID4 (regulated gene expression), DUOXA2 (ER protein), and PLIN2 (coats intracellular lipid storage) are all upregulated under CM stress.”

[Reviewer]

3. Line 303, “the list of binding partners.” Where is this list coming from? What binding partners? Interacting proteins or TF/co-factors that occupy the gene locus? This is not clearly stated.

[Authors]

To clarify this in the text, we now say:

“CIB1 has been shown to interact with a multitude of proteins⁵³. When CIB1 binds with PRKDC, TBPL1, PSEN2, PPP3R1, PTK2, PAK1, PDK1, KCNN1, or ITGA2B it activates these proteins, and these genes have increased expression under BFA treated β -cells.

However, the gene expression of proteins that CIB1 inhibits, are higher in TG-treated cells (Fig. 5d)."

To further provide clarification, the list of binding partners is from Leisner et al (DOI: [10.1096/fj.201500073R](https://doi.org/10.1096/fj.201500073R)). This is a list of CIB1-interacting proteins from additional publications.

REVIEWER COMMENTS

Reviewer #1 (Remarks to the Author):

The authors have adequately replied to all my comments. I would prefer if the western blot images were included in extended figure 9 together with the quantification to have a better presentation of the data.

Reviewer #2 (Remarks to the Author):

The authors have addressed some though not all issues raised in prior review. Suitability for publication in Nature Communication is up to the Editors.

Reviewer #3 (Remarks to the Author):

In this revision, the authors have significantly improved the quality and addressed the majority of concerns. I have a few additional minor comments here:

1. The authors included a large number of scRNA-seq from additional fixed samples, which were integrated with the original analysis. However, this integration method was not described in the methods.
2. In Fig 3h, when comparing the expression of fixed and hashed samples, beta and delta cells have very high correlations (0.88 and 1), whereas the correlation between alpha cells is much lower (0.61). Do the authors have an explanation for this discrepancy?
3. Fig 3h, ATAC signals for the ductal specific enhancer for CFTR drop dramatically in cytokine mix(CM) treatment. Does this decrease also affect the gene expression of CFTR?
4. In Figure 4a, are there any principle differences between "shared" stress responses and beta cell-specific stress response genes? It would be nice to show the gene ontology of alpha-beta shared genes vs. beta-only genes in BFA treatment (or CM).
5. Fig 6c, why do all other cell cycle genes show differential expression but Mki67 show no changes at all (completely white suggesting z-score is zero for all conditions)?

REVIEWER COMMENTS

[Reviewers comments in black]

[Author comments in dark blue]

[Reviewer]

Reviewer #1 (Remarks to the Author):

The authors have adequately replied to all my comments. I would prefer if the western blot images were included in extended figure 9 together with the quantification to have a better presentation of the data.

[Authors]

We again really appreciate the feedback from this reviewer and feel that we have a much better manuscript because of it.

We agree that this will improve the presentation of the data. We have added these blots to the Extended Figure 9.

[Reviewers comments in black]

[Author comments in dark blue]

Reviewer #2 (Remarks to the Author):

The authors have addressed some though not all issues raised in prior review. Suitability for publication in Nature Communication is up to the Editors.

[Authors]

We again really appreciate the feedback from this reviewer and feel that we have a much better manuscript because of it.

[Reviewers comments in black]

[Author comments in dark blue]

Reviewer #3 (Remarks to the Author):

In this revision, the authors have significantly improved the quality and addressed the majority of concerns. I have a few additional minor comments here:

[Authors]

We again really appreciate the feedback from this reviewer and feel that we have a much better manuscript because of it. We have addressed each of your additional minor comments below.

1. The authors included a large number of scRNA-seq from additional fixed samples, which were integrated with the original analysis. However, this integration method was not described in the methods.

[Authors]

We have added additional details in the methods section for the fixed samples and integration, now saying:

“For the fixed samples, the different stress treatments were merged using the raw data counts. Then, patients 4 and 5 were filtered for 200-9000 genes in each cell and percent mitochondria below 5. SCTransform was used to normalize each patient and regressed out percent.mt. “RunUMAP”, “FindNeighbors”, and “FindClusters” were done on each patient. The dimensions of patient 4 clustering and dimensional reduction were 1:25 and a resolution of 1.6. The dimensions of patient 5 were 1:35 and a resolution of 1.6. To integrate the two fixed data sets, we used “SelectIntegrationFeatures,” with features set to 3000. Then, we used “FindIntegrationAnchors” and the integration features to find anchors. Next, “IntegrateData” was used to integrate data from the anchors. For UMAP projection, “RunUMAP” and “FindNeighbors” were set to a dimension of 1:25, and “FindClusters” at a resolution of 1.6.

To integrate fixed and hashed sequencing, each individual patient was Lognormalized using “NormalizeData”. Then “SelectIntegrationFeatures”, “FindIntegrationAnchors”, and “IntegrateData” was performed. The data was scaled, and “RunPCA” was performed. The dimensions for “RunUMAP” and “FindNeighbors” were set to 1:27, and “FindClusters” had a resolution of 1.5. The integration of fixed and hashed samples was used for Extended Data Fig 3.”

[Reviewer]

2. In Fig 3h, when comparing the expression of fixed and hashed samples, beta and delta cells have very high correlations (0.88 and 1), whereas the correlation between alpha cells is much lower (0.61). Do the authors have an explanation for this discrepancy?

[Authors]

We have clarified this topic in the text, now stating:

“We also conducted a Pearson correlation between hashed CTRL and fixed CTRL samples to confirm cell type populations between sequencing methods. We found the highest correlation between the same cell types across sequencing methods (Extended Data Fig. 3g-h). We also observed variation in the correlation value of matched cell types across cell types, which may be due to donor and processing variations. These data are consistent with previous reports of these exogenous stressors^{15,32} and validate our scRNAseq approach to studying human islets cultured under defined chemical stressors.”

In addition, the main finding from Extended Data Fig 3h is that the cell types identified in each type of sequencing method had the highest correlation with the corresponding cell type in the other sequencing method. Even though alpha cells have a lower correlation than beta or delta cells, alpha-hashed cells have the highest correlation to alpha-fixed cells compared to other cell types, supporting the use of these methods in our study.

[Reviewer]

3. Fig 3h, ATAC signals for the ductal specific enhancer for CFTR drop dramatically in cytokine mix(CM) treatment. Does this decrease also affect the gene expression of CFTR?

[Authors]

We have performed this additional analysis and indeed see a significant decrease in CFTR gene expression in the duct with cytokine mix treatment, as we would expect from the ATAC data. We have updated Figure 3h to include a violin plot showing CFTR RNA expression levels, and this data can also now be found in Supplementary Data 8. In the text, we now say:

“There was also a significant decrease in RNA expression of CFTR under CM treatment in ductal cells (Fig. 3h, Supplementary Data 8).”

[Reviewer]

4. In Figure 4a, are there any principle differences between "shared" stress responses and beta cell-specific stress response genes? It would be nice to show the gene ontology of alpha-beta shared genes vs. beta-only genes in BFA treatment (or CM).

[Authors]

We have performed additional analysis using EnrichR on the differentially expressed genes in the Venn diagrams in Figure 4a to gain insights on shared and beta cell-only gene ontology. We conducted gene ontology using GO Biological Process 2023, GO Cellular Component 2023, and GO Molecular Function 2023 and have added this data to Supplementary Table 11.

This additional analysis revealed several new insights. We have included the following several pieces of text in the manuscript sharing each of these insights, now saying:

“Gene ontology analysis of shared genes under BFA stress revealed pathways related to ER-golgi (GO:0006888, GO:0048193, GO:0030134, GO:0012507, GO:0030127), which was expected due to the mechanisms of stress induced by BFA (Supplementary Table 11).”

Several genes associated with endocrine identity and insulin secretion (GO:0050796) were among the 286 downregulated genes shared between α - and β -cells.”

“Upregulated genes shared between α -, β -, and δ -cells under CM stress are related to inflammation (GO:0050729, GO:0006954, GO:0050727) and immune response (GO:0034341, GO:0045088), while downregulated genes are associated with endocrine identity and ribosomes (GO:0042254, GO:0042274, GO:0042255) (Supplementary Table 11).”

“Interestingly, gene ontology showed only two pathways specifically upregulated in α -cells and not β -cells; Protein Serine/Threonine Phosphatase Activity (GO:0004722) has previously been implicated in other models of diabetes⁴² and Vesicle (GO:0031982).”

“Gene ontology of upregulated genes in β -cells are related to double-stranded RNA binding. Recent data has indicated disruptions to RNA editing can upregulate double-stranded RNA leading to increases in inflammatory response and islet cell death⁴³.”

“Only one significant gene ontology pathway, Transition Metal Ion Binding (GO:0046914), was related to genes upregulated under CM stress in β -cells compared to α - or δ -cells (Supplementary Table 11).”

“Downregulated genes in β -cells are associated with metabolic pathways (GO:0031966, GO:0005747, GO:0005761, GO:0045333). Metabolism is essential for the proper function of pancreatic β -cells, and metabolic changes could result in differences in post-translational modifications related to T1D⁴³.”

“In addition, gene ontology terms related to tight junctions (GO:0070160) and Cadherin Binding (GO:0045296) are downregulated in α -cells under CM stress. This could implicate a disruption to cell-to-cell contact during CM stress.”

[Reviewer]

5. Fig 6c, why do all other cell cycle genes show differential expression but Mki67 show no changes at all (completely white suggesting z-score is zero for all conditions)?

[Authors]

We have included a better explanation in the text, now saying:

“We also found that BFA induces upregulation of proliferation and cell cycle markers⁵⁶ in β - and α -cells, however, MKI67 had the same expression across all conditions (Fig. 6c, Extended Data Fig. 12a). We are uncertain how much the lack of MKI67 variation is caused by technical or biological factors at this time.”

Despite the lack of variation in MKI67 expression, we decided to include it in Figure 6 because of how often this marker is looked at in the context of cellular proliferation. We do not know why we observe this but have some possibilities. This could be a technical limitation of scRNA-seq with low MKI67 expression, as MKI67 (which encodes the protein KI67), is typically studied with histological methods. There could be cell cycle-dependent

expression or different roles of each of these proliferation markers that would make some markers change while MKI67 does not.

REVIEWER COMMENTS

Reviewer #1 (Remarks to the Author):

The authors have adequately replied to all my comments. I would prefer if the western blot images were included in extended figure 9 together with the quantification to have a better presentation of the data.

Reviewer #2 (Remarks to the Author):

The authors have addressed some though not all issues raised in prior review. Suitability for publication in Nature Communication is up to the Editors.

REVIEWERS' COMMENTS

Reviewer #3 (Remarks to the Author):

The authors have addressed all my concerns. I believe this manuscript is now ready for publication.